# Using surface drifters to characterise near-surface ocean dynamics in the southern North Sea: a data-driven approach

Jimena Medina-Rubio[1], Madlene Nussbaum[2], Ton van den Bremer[3], and Erik van Sebille[1]

[1]Department of Physics, Institute for Marine and Atmospheric Research (IMAU), Utrecht University, Utrecht, The Netherlands
[2]Faculty of Geosciences, Physical Geography, Utrecht University, Utrecht, The Netherlands
[3]Faculty of Civil Engineering and Geosciences, Delft University of Technology, Delft, The Netherlands

**Correspondence:** Jimena Medina-Rubio (j.medinarubio@uu.nl)

**Abstract.** The large size of traditional drifters limits their ability to mimic the transport of buoyant objects at the ocean surface, which are subject to complex interactions among direct wind drag, fast-moving surface currents, and wave-induced transport. To better capture these dynamics, we track the trajectories of 12 novel, ultra-thin surface drifters deployed in the southern North Sea over 68 days. We adopt a data-driven approach to model drifter velocity using hydrodynamic and atmospheric data, applying both a linear leeway parameterisation and two machine learning models: random forest and support vector regression. Machine learning model-agnostic interpretation techniques reveal that tidal forcing predominantly drives zonal motion, whereas wind is the main driver in the meridional direction in this region. Notably, the wind exhibits a saturation effect, and its contribution to explaining the variance of the drifter velocity decreases at higher speeds. In trajectory prediction experiments, we find that machine learning models, particularly random forest, outperform linear models, with the latter achieving comparable accuracy only at short time scales. Using a hybrid approach and deriving a non-linear function of the wind from machine learning interpretable methods to include in the leeway parameterisation significantly improves the model prediction of the drifter trajectory. Finally, we test the generalisability of the North Sea–trained models using an independent drifter dataset from the Tyrrhenian Sea. Despite the differences in ocean dynamics between the regions, the machine learning models reproduce the observed trajectories with comparable accuracy to state-of-the-art studies, demonstrating robust explanatory skill and a low degree of overfitting in this instance.

## 1 Introduction

Accurate predictions of the pathways and the fate of buoyant objects in the ocean rely on our understanding of surface ocean dynamics (Röhrs et al., 2021). For example, search-and-rescue missions require a model that predicts how far a missing person has drifted at the ocean surface to define the search area (Breivik et al., 2013). These models also benefit biological studies, where the degree of connectivity between different oceanic regions affects the population dynamics of species such as phytoplankton, larvae, and turtles (Nooteboom et al., 2019; Grimaldi et al., 2022; Lindo-Atichati et al., 2020; Manral et al., 2024). Similarly, oil spill detection and modelling depend on our knowledge of geophysical fluid dynamics to mitigate environmental damage (Pisano et al., 2016; Jones et al., 2016; Calzada et al., 2021). Another pressing concern is plastic

pollution, which threatens marine fauna through ingestion and entanglement (Kühn and van Franeker, 2020), and disrupts
ecosystems via chemical contamination (Mato et al., 2001) and the facilitation of biological invasions (Haram et al., 2023).
Marine debris has increased by 4% on average every year since 1980, indicating that the current plastic standing stock of 3,200
kilotons in the ocean will double within two decades, exacerbating the issue (Kaandorp et al., 2023). To further advance these
areas of operational oceanography and improve our fundamental understanding of near-surface ocean dynamics, modelling the
physical mechanisms driving the transport of buoyant objects in the ocean is key.

The transport of buoyant objects at the ocean surface is governed by a complex interplay of winds, waves, and ocean currents
at different spatio-temporal scales. Wind stress creates a friction force on floating objects and generates near-surface currents,
resulting in a strong vertical velocity shear, following the well-known steady-state Ekman solution (Ekman, 1905). Surface
gravity waves contribute to transport in the direction of wave propagation through Stokes drift, which arises from deviations
in orbital motion (Stokes, 1847). The interaction between Stokes drift and the Coriolis force generates the Coriolis–Stokes
force, which accelerates a current perpendicular to the Stokes drift (see van den Bremer and Breivik (2018)). In addition,
buoyant objects are transported by low-frequency currents, including geostrophic flows that drive large-scale ocean circulation,
topographic steering, mesoscale eddies, and high-frequency periodic currents such as tides and inertial oscillations (Röhrs
et al., 2021).

Buoyant drifters can be used to monitor how these different physical mechanisms combine to drive the transport at the
ocean near-surface (Lumpkin et al., 2016). Currently, the most commonly deployed drifters are transported with a current
that is effectively integrated over part of the water column (over the vertical extent of the drifter) and do not capture directly
the complex surface ocean dynamics (Davis et al., 1982; Sybrandy and Niiler, 1992; Novelli et al., 2017; MetOcean, 2017).
Alternatively, the surface drifters used in this study (see MetOcean (2020)) have a thin disc shape with a height of $4.1\,\mathrm{cm}$ that
enables them to follow the orbital velocities of the waves and drift with the uppermost centimetres of the ocean surface currents
(Morey et al., 2018; Calvert et al., 2024; Pawlowicz et al., 2024), which are characterised by higher speeds. Yet, the relevance
of the Stokes drift, wind drag, or surface currents on these specific drifters is currently unclear.

Through the analysis of the trajectories of our disc-shaped surface drifters and their comparison with model simulations, we
can estimate how each forcing mechanism contributes to the near-surface ocean transport. A prevalent data-driven methodology
for quantifying these contributions involves regression models, which seek to establish a relationship between drifter velocity
observations and the hydrodynamic and atmospheric conditions. As described by Breiman (2001a), these regression models
that aim to describe natural processes fall into two broad categories: traditional statistical models (also known as linear models,
Bishop (2006)) or algorithmic models, also known as supervised Machine Learning (ML). Linear models require manually
defining the model structure to approximate non-linearities and interdependencies of the near-surface ocean dynamics. This
poses a challenge, as there are high-dimensional processes in the ocean with spatiotemporal scales spanning several orders
of magnitude and involving many degrees of freedom (van Sebille et al., 2020). In contrast, machine learning regression
methods do not require explicit knowledge of the underlying transport mechanisms as they establish relationships driven by the
algorithms, making them particularly useful for complex systems with multiscale variability and non-linear interdependence
(Bracco et al., 2025). For instance, Callies et al. (2017) demonstrated a consistent relationship between the leeway (i.e., wind

contribution to surface friction) and Stokes drift over time in the same region, illustrating these interdependencies of the system. Furthermore, previous studies have highlighted the advantages in the prediction of the fate of buoyant objects at the ocean surface using different machine learning methods, such as tree-based models (Kaandorp et al., 2022; O'Malley et al., 2023) and neural networks (Aksamit et al., 2020; Fajardo-Urbina et al., 2024; Grossi et al., 2025).

In this study, we analyse the trajectories of twelve disc-shaped surface drifters deployed in the southern North Sea to evaluate how different forcing mechanisms drive changes in their velocity, using data on the surface ocean currents, wave conditions, and wind patterns. We apply a widely used linear parametrisation and two machine learning regression algorithms (random forest and support vector regression) to model the velocity of the drifters. Using this data-driven approach, we aim to (i) identify the dominant forces driving drifter velocity changes and (ii) improve the prediction of drifter trajectories using only physical variables describing near-surface ocean dynamics.

## 2 Data

### 2.1 Surface drifters

#### 2.1.1 Design and deployment

The drifters are thin disc-shaped buoys manufactured by MetOcean (MetOcean, 2020). Each drifter has a diameter of $24\,\mathrm{cm}$ a height of $4.1\,\mathrm{cm}$, and a weight of $900\,\mathrm{g}$ . These drifters are designed to track the uppermost centimetres of the ocean surface. They are equipped with Global Navigation Satellite System (GNSS) positioning ($8\,\mathrm{m}$ positional accuracy), a sea surface temperature (SST) sensor, and Iridium satellite telemetry. Details on spatial coordinate error estimation are provided in Appendix A. To facilitate the retrieval of beached drifters after battery depletion, we attached AirTags (Apple Inc., 2021), increasing the total mass by $1\%$. Additionally, the drifters report which of their two antennas is transmitting, distinguishing between the upward-facing and submerged positions. This allows us to monitor their orientation and thus their flipping behaviour.

We released the drifters off the coast of the town of Moddergat, the Netherlands, in the Eastern Wadden Sea on 25 April 2024 (Fig. 1). We placed the drifters in three clusters spaced $250\,\mathrm{m}$ apart on mudflats at low tide, awaiting the incoming tide to transport them. Within each cluster, we arranged four drifters in a square formation with $50\,\mathrm{m}$ spacing between them. All drifters remained operational for at least 68 days, drifting across the German Bight region in the southern North Sea and reaching the North Frisian Islands. Initially, we set the drifters' sampling frequency to five minutes, but we lengthened the interval to thirty minutes after six days to extend battery life. After 26 days, we further lengthened it to three hours.

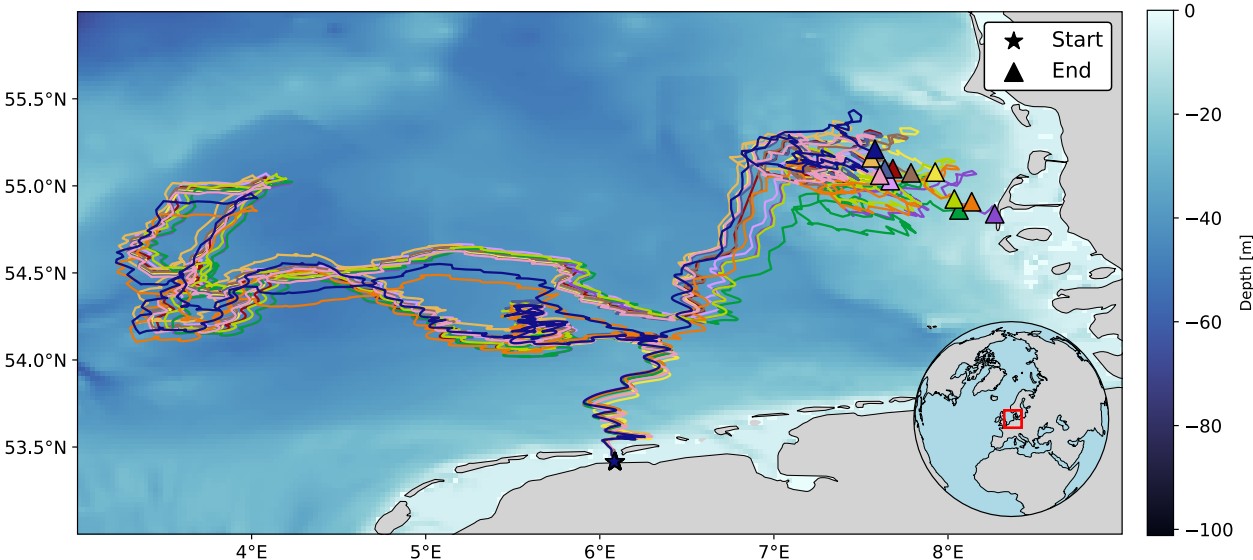

**Figure 1.** Trajectories over 68 days of 12 colour-coded surface drifters in the southern North Sea. Drifters were deployed on 25 April 2024 in the Dutch Eastern Wadden Sea in three different clusters spaced $250\,\text{m}$ apart. Starting and ending positions are marked with stars and triangles, respectively. Background colourmap shows the bathymetry of the southern North Sea from the NWS Ocean Physics Analysis and Forecast model with a horizontal resolution of $0.027°$ (Tonani et al., 2019). The study site location in the southern North Sea is highlighted by a red rectangle on the orthogonal projection of the Northern Hemisphere in the bottom right corner.

### 2.1.2 Drifter data processing

To ensure the quality of the GNSS data, data points for which the time difference between measurements $\Delta t$ is less than $2.5\,\text{min}$ are eliminated. This threshold removes redundant measurements, which typically result from minor time synchronisation errors between the drifter's internal time and the satellite time stamp or re-transmissions occurring within the intended sampling
interval. Furthermore, the atmospheric and ocean datasets to be used for comparison (Sect. 2.2) are only available on a much coarser time resolution. We compute the (total) drifter velocity $\boldsymbol{v}_d$ at position $\boldsymbol{x}_n$ and time $t_n$, corresponding to the n-th observation, using the central difference scheme, which averages the arriving and departing instantaneous velocities (Elipot et al., 2016). This vector is calculated as

$$\boldsymbol{v}_d(\boldsymbol{x}_n, t_n) = \frac{1}{2}\left(\frac{\boldsymbol{x}_{n+1} - \boldsymbol{x}_n}{t_{n+1} - t_n} + \frac{\boldsymbol{x}_n - \boldsymbol{x}_{n-1}}{t_n - t_{n-1}}\right) \tag{1}$$

This method yields average speeds for the entire data set of $0.27 \pm 0.19\,\text{ms}^{-1}$ in the zonal direction and $0.13 \pm 0.10\,\text{ms}^{-1}$ in the meridional direction. Measurements with speeds exceeding $3\,\text{ms}^{-1}$ are flagged as spatial coordinate errors, as sustained speeds above this threshold are considered physically unrealistic given the typical fluid motion timescales in the region (Otto et al., 1990). While such high velocities may occur during wave breaking, these events would last less than a fraction of $\Delta t$. In total, $0.6\%$ of the original data points are identified as signal recording errors and removed from the time series. We also omit

the first $24\,\mathrm{h}$ to reduce the importance of coastal and intertidal effects, focusing the analysis on mesoscale surface dynamics in the open basin.

The German Bight is characterised by strong tidal dynamics due to its shallowness (Otto et al., 1990), so to isolate the residual kinematics from these dominant tidal effects, we estimate the net drifter velocity $\boldsymbol{v}_d$ over the dominant tidal cycle $T$. This residual (Lagrangian) velocity is defined by Zimmerman (1979) as:

$$\tilde{\boldsymbol{v}}_d(\boldsymbol{x},t) = \frac{1}{T}\int_{t-\frac{T}{2}}^{t+\frac{T}{2}} \boldsymbol{v}_d(\boldsymbol{x},t')\mathrm{d}t' \tag{2}$$

We identify the dominant tidal harmonic in the drifters' velocity data using Fast Fourier Transform (FFT) and Morlet wavelet analysis (Meyers et al., 1993). Both methods show that the semi-diurnal $M_2$ and $S_2$ tidal constituents are the most significant, with no signal detected at the inertial period. These findings are consistent with prior Lagrangian observations in the region (Meyerjürgens et al., 2019; Deyle et al., 2024). Hence, drifter velocities are time-averaged over the period $T = 24.83\,\mathrm{h}$ to
smooth out the influence of both tidal signals, resulting in residual speeds that reach a maximum of $0.63\,\mathrm{m\,s^{-1}}$. Further details on the spectral analysis methodology and a visualisation of the frequency spectra can be found in Appendix B.

## 2.2  Atmospheric and hydrodynamic datasets

We use an atmospheric model and coupled ocean–wave models to quantify the effects of the different forcing mechanisms on drifter transport. We use data of the wind velocity field at $10\,\mathrm{m}$ above the ocean surface ($\boldsymbol{u}_w$) from the EMCWF reanalysis
model ERA5, which assimilates satellite and in-situ measurements (Hersbach et al., 2023). The surface ocean currents velocity field ($\boldsymbol{u}_o$) is provided by the North Western Shelf Ocean Physics Analysis and Forecast model, which is forced by the EMCWF wind field and assimilates SST, sea level anomalies from satellites, as well as in situ temperature and salinity profiles (Copernicus Marine Service, 2024a). The model uses a hybrid $z^* - \sigma$ terrain-following vertical coordinate system consisting of 51 levels, with the thickness of the surface cell set to $\leq 1\,\mathrm{m}$ (Tonani et al., 2019). Due to the strong tidal ocean dynamics
in the region, we isolate tidal contributions from the ocean surface velocity field using a low-pass filter with a time window $T = 24.83\,\mathrm{h}$. The resulting low-pass ocean currents velocity vector is denoted as $\boldsymbol{u}_o^{LP}$, while the remaining high-pass filtered ocean currents velocity will be referred to as $\boldsymbol{u}_o^{HP}$. To characterise the wave conditions at the southern North Sea, we use the NWS Ocean-Wave Forecasting System (Copernicus Marine Service, 2024b), which is coupled to the mentioned ocean currents model and includes the effect of wave-induced fluxes in momentum and energy, and the Coriolis–Stokes force on the Eulerian
current. Several studies have emphasised the importance of using coupled models (compared to non-coupled models) in the simulation of wave-induced surface transport of buoyant objects (Röhrs et al., 2012; Cunningham et al., 2022; Rühs et al., 2025). Three key parameters used to describe the effect of ocean waves on the transport of buoyant objects are significant wave height, mean propagation direction, and mean frequency or period. These parameters are derived from the wave spectrum model output, which partitions the spectral significant wave height ($H_s$) and the mean wave direction ($\theta$) into wind waves, and first
and second swell components. The mean wave direction represents the energy-weighted average propagation direction within

each spectral partition. In contrast, the bulk wave direction ($\theta^{\text{bulk}}$) is computed from the full, unpartitioned directional spectrum and represents the overall propagation direction of the total wave field. Additionally, we consider the period at the spectral peak ($T_p$), and the Stokes drift velocity field at the surface ($\boldsymbol{u}_s$). As in Bruciaferri et al. (2021), we calculate the Stokes drift at $0.5\,\text{m}$ below the still–water level to align with the depth mid-point of the upper ocean model layer, and apply the parametrisation of Breivik et al. (2016) based on the Phillips wind-wave spectrum. While this approximation may underestimate the Stokes drift experienced by the drifters (Lenain and Pizzo, 2020), defining the Stokes drift at this depth ensures consistency, allowing a coherent analysis of the effects derived from the coupled ocean–wave model. Further details on the models' spatio-temporal resolution are presented in Table 1.

**Table 1.** Specifications of the hydrodynamic and atmospheric model data at the drifters' spatiotemporal coordinates. Variables included are the wind velocity vector ($\boldsymbol{u}_w$), high-pass and low-pass ocean currents velocity using a $24.83\,\text{h}$ filter ($\boldsymbol{u}_o^{HP}$, $\boldsymbol{u}_o^{LP}$), Stokes drift velocity in the ocean upper-layer ($\boldsymbol{u}_s$), and properties derived from the wave spectrum, including significant wave height from the wind, first and second swell partitions ($H_s^{\text{wind}}, H_s^{\text{1p swell}}, H_s^{\text{2p swell}}$), wave direction from the wind, and first and second swell partitions ($\theta^{\text{wind}}, \theta^{\text{1p swell}}, \theta^{\text{2p swell}}$), bulk wave direction ($\theta^{\text{bulk}}$), and wave period at the spectral peak ($T_p$).

| Variables | Dataset | Spatial res. | Temporal res. | Ref. |
|---|---|---|---|---|
| $\boldsymbol{u}_w$ | ERA5 global reanalysis | $0.25° \times 0.25°$ | $1\,\text{h}$ | Hersbach et al. (2023) |
| $\boldsymbol{u}_o^{HP}, \boldsymbol{u}_o^{LP}$ | NWS Ocean Physics Analysis and Forecast | $0.027° \times 0.027°$ | $15\,\text{min}$ instantaneous | Tonani et al. (2019) |
| $H_s^{\text{wind}}, H_s^{\text{1p swell}}, H_s^{\text{2p swell}}, \theta^{\text{wind}}, \theta^{\text{1p swell}}, \theta^{\text{2p swell}}, \theta^{\text{bulk}}, T_p, \boldsymbol{u}_s$ | NWS Ocean-Wave Forecasting System | $0.05° \times 0.05°$ | $1\,\text{h}$ | Bruciaferri et al. (2021) |

Where needed, we interpolate the atmospheric and hydrodynamic model data to the drifter measurements of location and timestamp to assess their instantaneous effect on their velocity using the bilinear interpolation scheme from Parcels (Delandmeter and van Sebille, 2019). To align with the models' $1\,\text{h}$ temporal resolution while reducing high-frequency noise, drifter coordinates during the period with $\Delta t = 5\,\text{min}$ are resampled to a coarser resolution of $30\,\text{min}$ via linear interpolation.

## 3 Methodology

We manipulate the interpolated hydrodynamic and atmospheric model data to define physically relevant independent variables to model velocity changes along the trajectories of the drifters. We include the zonal ($U$) and meridional ($V$) components of all velocity vectors, including the surface ocean currents, wind, and Stokes drift. To account for wave directionality, we project the properties derived from the wave spectrum onto the zonal and meridional axes using wave direction data. Let $A$ be one such property; we formulate its vector form by $\boldsymbol{A} = (A\sin\theta, A\cos\theta)$, where $\theta$ is the deviation angle from true north of the wave direction. We define the peak spectral period vector $\boldsymbol{T}_p = (T_p^x, T_p^y) = (T_p\sin\theta^{\text{bulk}}, T_p\cos\theta^{\text{bulk}})$ using the bulk wave direction $\theta^{\text{bulk}}$ and, in doing so, assign a scalar to the two directions in an ad hoc fashion. Similarly, we compute three different

significant wave height vectors, each associated with a specific wave partition (wind sea, first swell, and second swell). These are expressed as $\boldsymbol{H}_s^i = (H_s^{i,x}, H_s^{i,y}) = (H_s^i \sin\theta^i, H_s^i \cos\theta^i)$, where $i$ denotes the wave partition, and each vector is calculated using the corresponding wave direction for that partition. This ensures that the models incorporate wave effects on drifter motion in both directions, resulting in a feature matrix comprised of 16 variables measured over $18,696$ time points. We do not include previous history of hydrodynamic and atmospheric conditions (i.e., we do not include variables with time lags or averages over a spatial radius of influence) because these variables would physically represent an inertial effect on the drifters, which has been found to be very small (Olascoaga et al., 2020).

To infer the predominant forcing mechanisms and model the trajectory of the drifters, we analyse the outcome using the baseline linear regression model of the total drifter velocity components and compare it to two non-linear machine learning algorithms: random forest and support vector regression. Given the study region's stronger zonal than meridional tidal component (Kopte et al., 2022), we model the zonal ($U_d$) and meridional ($V_d$) drifter velocity components separately in both the linear and machine learning models and assume that the interdependence between these two variables is negligible.

### 3.1 Linear regression

Unlike more complex non-linear models, linear regression offers a direct model interpretation by quantifying each feature's (i.e., independent variables) influence through interpretable coefficients or weights in the prediction equation of the target (i.e., dependent variable). The total drifter velocity $\boldsymbol{u}_d$ is typically parameterised using a physics-based linear model as

$$\boldsymbol{u}_d = \boldsymbol{u}_o + \boldsymbol{u}_s + \boldsymbol{\gamma} \odot \boldsymbol{u}_w + \varepsilon, \tag{3}$$

where $\boldsymbol{\gamma}$ is a vector of model coefficients that scales each component of the wind velocity $\boldsymbol{u}_w$, $\varepsilon$ is a disturbance term to be minimised by the linear regression algorithm, $\boldsymbol{u}_o$ is the surface currents total velocity (i.e., the addition of the low-pass and high-pass surface ocean currents), $\boldsymbol{u}_s$ is the Stokes drift velocity, and $\odot$ is the Hadamard or element-wise product. Note that this vector equation represents two independent scalar equations, one for the zonal and one for the meridional components.

Physically, the wind term weight represents the drag coefficient ratio above and below water, since our drifters are radially and axially symmetric (Dominicis et al., 2016). Yet, downwind and crosswind components of the drifter velocity vector have been found to have distinct dependences on the wind speed (Allen, 2005). Several studies have hence divided the wind contribution into two perpendicular components, known in the literature as the "leeway method" (Breivik et al., 2011), and quantified the wind slip vector represented by $\boldsymbol{\gamma}$ in Eq. (3). This approach yields values of $\boldsymbol{\gamma}$ ranging from $1-3\%$, depending on the type of surface drifters and choice of hydrodynamic model (Sutherland et al., 2020; Staneva et al., 2021). Hence, we treat wind slip $\boldsymbol{\gamma} = (\gamma^x, \gamma^y)$ as a vector with zonal and meridional components to find the best-fit values for this new drifter design using the ordinary least squares method (Faraway, 2005). The accuracy of the linear model is evaluated based on its goodness-of-fit. Although this method is an empirical parametrisation of the wind contribution to the drifter velocity, its functional form aligns with theoretically derived models of the drift of spherical buoyant objects at the ocean surface (Beron-Vera et al., 2019).

We also consider a linear regression model in terms of the relative wind, given that the friction velocity (i.e., the actual velocity acting at the ocean surface) is a function of the wind speed with a slope equal to the drag coefficient and a non-zero

constant (Foreman and Emeis, 2010). Upon fitting the boundary conditions at the ocean surface, the drifter velocity is hence expressed as

$$\boldsymbol{u}_d = \boldsymbol{u}_o + \boldsymbol{u}_s + \boldsymbol{\gamma} \odot (\boldsymbol{u}_w - \boldsymbol{u}_o) + \varepsilon \tag{4}$$

Making assumptions on the magnitude and spatio-temporal scales of the wind and ocean current forcing simplifies this formulation back to Eq. (3) as shown by Duhaut and Straub (2006).

Nevertheless, despite the high interpretability of linear models, they may still oversimplify near-surface ocean dynamics by omitting non-linear behaviour in the parameterisation of drifter velocity components. For the case of highly correlated features, linear models also struggle to determine their contributions, leading to instability in coefficient estimation and reduced model reliability (Molnar, 2022). From a physical perspective, another limitation is that the surface current velocities used here are depth averaged over the model's upper layer (Tonani et al., 2019). This means that wind-driven vertical shear in the upper centimetres may be underestimated and part of the shear effect effectively absorbed by the windage coefficient, potentially influencing the interpretation of the modelled surface currents (Callies et al., 2017; Laxague et al., 2018).

## 3.2 Machine learning regression

Machine learning algorithms offer an alternative regression approach particularly suited to climate variables, capturing non-linear interactions and multicollinearity without requiring prior structural knowledge (Breiman, 2001b). We create a random forest model, as it has been found to perform well for spatio-temporal predictions due to its capacity to handle highly correlated features (Hengl et al., 2018; Nussbaum et al., 2018). To contrast these results, we also employ a support vector regression model and take advantage of its ability to handle high-dimensional feature spaces and its intrinsic algorithmic differences with the decision-tree structure of the random forest model. Both models are implemented using the *scikit-learn* Python package (Pedregosa et al. (2011), version 1.6.1).

### 3.2.1 Random forest

Random forest is a machine learning method that predicts the values of the target variable from the average of the predictions from a collection of decision trees (Breiman, 2001b). Each regression tree recursively partitions the data through binary splits based on feature thresholds. In this algorithm, each tree is fitted to a randomly drawn subset of the data and uses a randomly drawn subset of features to consider at each split. Single decision trees struggle with linear relationships, which must be approximated by step functions, and are sensitive to small input changes, sometimes producing non-smooth predictions (Molnar, 2022). By averaging over many trees, random forests reduce these limitations, exhibiting low sensitivity to hyperparameter choices including the number of trees $n_{\text{tree}}$, the minimal number of observations at terminal nodes $n_{\text{min}}$ (i.e., the last branch of each tree), and the number of randomly selected features to test at each split $m_{\text{try}}$ (Probst and Boulesteix, 2017). Hence, we build a random forest model to fit the drifter velocity zonal and meridional components with *scikit-learn* defaults $n_{\text{tree}} = 100$, $n_{\text{min}} = 2$, $m_{\text{try}} = n_p$ (Geurts et al., 2006) where $n_p$ is the total number of variables in the feature matrix (sometimes called predictors) and $n_p = 16$.

### 3.2.2 Support vector regression

Support vector regression is a model that applies principles of the support vector machine (Cortes and Vapnik, 1995) to regression tasks (Drucker et al., 1996). In classification, support vector machines find the optimal hyperplane that separates data and maximises its distance to the closest data point (Vapnik, 1999). Instead, support vector regression identifies an optimal hyperplane with a margin (defined by *support vectors*) where prediction errors are tolerated. This model applies a transformation using non-linear kernel functions to project the data into a higher-dimensional space where a linear hyperplane can better approximate the non-linear relationships within the data (Smola and Schölkopf, 2004). However, the support vector regression model is sensitive to the choice of kernel and hyperparameters, which can strongly affect model performance. To address this, we use a radial basis function (RBF) and optimise the hyperparameters via grid search using cross-validation (see Appendix D1 for the exact values).

### 3.2.3 Evaluation of predictive performance

We evaluate the predictive performance of the machine learning models through cross-validation, where we calculate the expected extra-sample error (Hastie et al., 2009). Validating the predictive performance of these models for drifter velocities is challenging due to the strong temporal autocorrelation present in trajectory data and the limited spatial dispersion in our dataset (Wadoux and Heuvelink, 2023). These factors could potentially lead to overly optimistic model performance estimates if training and testing sets are not sufficiently independent. To address this, we designed a spatio-temporal block cross-validation strategy to organise data into independent time blocks, regardless of the drifter, and then applied a k-fold cross-validation ($k = 5$) by selecting a random set of blocks to be left out in each fold. Doing so ensures that the model is validated on data that is both temporally and spatially independent of the training set. The duration of the blocks corresponds to the autocorrelation time of the target variables, ensuring that any two points separated by more than this time range can be considered statistically independent and thus suitable for validation. This approach substantially reduces the risk of overfitting, as it prevents information leakage between training and validation sets and ensures that model performance is assessed on truly unseen, time-independent samples. For a detailed explanation of the block-cross validation strategy and the exact autocorrelation times, see Appendix C.

The zonal and meridional drifter velocity models' predictive accuracy are evaluated independently using the coefficient of determination $R^2$ (Wilks, 2011), which quantifies the variance explained by the model, the root-mean-square error (RMSE), which measures the average prediction error magnitude, and the mean absolute error (MAE), which assesses the bias of the model. These metrics are defined as

$$R^2 = 1 - \frac{\sum_{i=1}^{N}(y_i - \hat{y}_i)^2}{\sum_{i=1}^{N}(y_i - \overline{y}_i)^2}, \qquad \text{RMSE} = \sqrt{\frac{1}{N}\sum_{i=1}^{N}(y_i - \hat{y}_i)^2}, \qquad \text{MAE} = \frac{1}{N}\sum_{i=1}^{N}|\,y_i - \hat{y}_i\,|, \tag{5}$$

where $y_i$ are the observed target values, $\hat{y}_i$ are the model predictions, $\overline{y}_i$ is the mean of the observed values, and $N$ is the number of observations. All metrics are applied for each fold using the observed and predicted data points from the test set

and averaged across folds to assess the models' predictive capacities. Cross-validation plots of predicted against observed data points within each testing fold are included in S1 in the Supplement.

### 3.2.4 Model interpretation

A key limitation of machine learning models is their reduced interpretability compared to linear models, the so-called "black-box" abstractness (Lipton, 2017). As these models do not use a functional form to calculate the output, it becomes hard to understand the effects of individual features on the final prediction of the target variable (Molnar, 2022). To mitigate this, model-agnostic methods have been developed to characterise the overall behaviour of machine learning models.

One such method is permutation feature importance, which is used for identifying the variables that have the greatest impact on the target. This approach quantifies the increase in model prediction error, chosen to be the RMSE, when the values of a feature are randomly shuffled 10 times along the time dimension (i.e., the value for observation $t_n$ is reassigned to $t_k$, where $k \neq n$ is a random index within the dataset). This assumes features causing large prediction error increases when shuffled are more important in explaining the variance in the target data (Ewald et al., 2024).

Another powerful tool for interpreting machine learning models post hoc are Accumulated Local Effects (ALE) plots. These ALE plots are model-agnostic methods for explaining individual predictions (Apley and Zhu, 2019). As described by Molnar (2022), ALE plots show how a model's predictions change with respect to a feature, considering only the data points within a local range (window) of that feature. To improve visualisation, the accumulated changes are computed across all such intervals covering the feature's range. An advantage of this method is its ability to visually reveal the functional relationship between the target variable and individual features. Unlike predecessors such as Partial Dependence Plots (PDPs) (Friedman, 2001), which estimate global average effects across the entire dataset and can be biased by correlated features, ALE plots capture local effects by focusing on small, localised changes in prediction within each interval. We estimate the ALE uncertainty by calculating the results from each feature using 100 bootstrapped resamples of the feature matrix and target variable. For each feature, we calculate the $95\%$ Confidence Interval (CI) based on the distribution of ALE estimates across these bootstrapped samples. These ALE plots are generated using the *PyAle* Python package (Jomar (2020), version 1.2.0), a Python implementation of the R package *ALEPlot* (Apley, 2018). After plotting the ALE, feature transformations are derived by visual inspection to approximate the functional form of the ALE curve to be able to obtain an interpretable model, i.e. the making the linear model non-linear but retaining its ease of interpretation properties.

### 3.3 Modelling drifter trajectories

To assess model predictive accuracy, we reconstruct the change in location of the drifters over time using the models' predictions of their total velocity and employing a leave-one-drifter-out cross-validation strategy. We train each of the three models on data from 11 drifters, constituting the training feature matrix $X$, and obtain the mapping functions $f_u(X)$ and $f_v(X)$ which relate the hydrodynamic and atmospheric ocean conditions to the total drifter velocity components. The predicted drifter velocity vector $\hat{\boldsymbol{u}}_d$ is expressed as:

$$\hat{\boldsymbol{u}}_d = \begin{pmatrix} \hat{U}_d \\ \hat{V}_d \end{pmatrix} = \begin{pmatrix} f_u(X) \\ f_v(X) \end{pmatrix},$$ (6)

where $\hat{U}_d$ and $\hat{V}_d$ are the zonal and meridional predicted drifter velocity components, respectively.

We simulate the trajectory of the test case, i.e., the excluded drifter, by integrating the velocity predictions over time. Let $X'(\boldsymbol{x}, t)$ contain interpolated hydrodynamic and atmospheric variables at position $\boldsymbol{x}_d$ and time $t$. The predicted drifter position $\hat{\boldsymbol{x}}_d(t)$, based on a given model, is defined as:

$$\hat{\boldsymbol{x}}_d(t + \delta t) = \hat{\boldsymbol{x}}_d(t) + \int\limits_{t}^{t+\delta t} \hat{\boldsymbol{u}}_d(X'(\hat{\boldsymbol{x}}_d, \tau)) \mathrm{d}\tau,$$ (7)

where dt is the integration time-step, set to $\mathrm{dt} = 60\,\mathrm{s}$. We use a forward Euler method to solve this integral and calculate the time advection of the drifter trajectories.

### 3.3.1 Trajectory prediction skill metrics

The accuracy of the modelled trajectories is measured using the mean cumulative separation distance $D$, which quantifies the difference between observed and modelled spatial coordinates at each time-step (Haza et al., 2019; van der Mheen et al., 2020; Moerman et al., 2024). This metric is calculated as

$$D = \frac{1}{M} \sum_{i=0}^{M} \|\hat{\boldsymbol{x}}_d(t_i) - \boldsymbol{x}_d(t_i)\|,$$ (8)

where $M$ is the total number of timesteps along the drifter trajectory, and $\hat{\boldsymbol{x}}_d(t_i)$ is the position vector of the drifter at the $i-$th timestep. As reference, observed drifters travel on average a total of $1,795\,\mathrm{km}$ over their measuring period.

We also evaluate the predicted trajectories using the Liu–Weisberg skill score (Liu and Weisberg, 2011), which is widely used in drifter studies (Liu et al., 2014; van Sebille et al., 2021; Pärn et al., 2023). This metric is the average of the separation distance (i.e., the distance between the model prediction and the observed position) weighted by the length of the observed trajectory. The skill score $ss$ is given by

$$ss = \begin{cases} 1 - \frac{s}{n}, & s \le n \\ 0, & s > n \end{cases} \quad \text{for} \quad s = \frac{\sum_{i=0}^{M} \|\hat{\boldsymbol{x}}_d(t_i) - \boldsymbol{x_d}(t_i)\|}{\sum_{i=0}^{M} \sum_{j=0}^{i} \|\boldsymbol{x}_d(t_{j+1}) - \boldsymbol{x}_d(t_j)\|},$$ (9)

where $n$ is a non-dimensional number that defines the threshold of no skill. Liu and Weisberg (2011) use a threshold of $n = 1$ to calculate the accuracy of 3-day predictions. However, due to the long prediction period in our trajectory analysis (up to 60 days), a smaller threshold value $n$ is needed. Otherwise, minor differences in separation distance between models would have little influence on the final Liu-Weisberg skill score, which would instead be dominated by the large cumulative trajectory

length. We therefore set the threshold to $n = 0.05$, preserving a consistent ratio between prediction period and threshold to ensure meaningful model comparisons.

### 3.3.2 Impact of non-dynamical variables on trajectory prediction

To draw physically meaningful conclusions about the dominant forces governing the transport of buoyant objects, our machine learning models are initially trained using only dynamic physical variables that change along the drifter trajectory and contrasted with the linear model. Nonetheless, prior studies in other fields have shown that including spatiotemporal features such as latitude, longitude, distance to reference points, time since release, or seasonal indicators can significantly enhance machine learning models' performance (Behrens et al., 2018; Hengl et al., 2018; O'Malley et al., 2023). To assess this in our data, we train an additional random forest model to predict total drifter velocity, incorporating non-dynamic features and contrast the results with the original random forest model. This random forest model with an expanded dataset includes features related to position (latitude and longitude) and local water depth, along with the original input variables. We also test whether explicitly accounting for the additional transport of these surface drifters due to possible wave surfing (Pizzo et al., 2019) improves the random forest model predictive skill by introducing another feature: Flipping Index. This feature parameterises drifter flipping behaviour, a process used by Haza et al. (2018) for the identification of drogue loss and associated with storm conditions with strong winds and high wave steepness that induce wave-breaking. The Flipping Index quantifies the proportion of orientation changes (i.e., flips) based on successive antenna measurements (see Appendix E for full details). In order to obtain a Flipping Index along the drifters' trajectories, we first train a random forest model to predict it everywhere. Since only about $28\%$ of the dataset exhibits non-zero Flipping Index values (Fig. E2), we apply a hurdle modelling approach: first, a random forest classifier predicts the likelihood of flipping, and second, a random forest regressor estimates the flipping magnitude conditional on a positive event (Wadoux and Heuvelink, 2023).

## 4 Results and Discussion

### 4.1 Inference of predominant forcing mechanisms

The linear regression, random forest, and support vector regression models collectively provide insight into the physical forcing mechanisms governing the transport of buoyant objects at the ocean surface. By analysing both the total and residual drifter velocities, we can infer the relative importance of wind, waves, and ocean currents in shaping measured trajectories of our surface drifters at subtidal and supertidal scales.

### 4.1.1 Total drifter velocity

Fitting the weights of the ordinary least-square linear regression given by Eq. (3) of the drifter total zonal and meridional velocity components yields $\gamma^x = 1.34\%$ ($R^2 = 0.26$, $\text{RMSE} = 0.11\,\text{ms}^{-1}$, $\text{MAE} = 0.12\,\text{ms}^{-1}$) and $\gamma^y = 1.63\%$ ($R^2 = 0.40$, $\text{RMSE} = 0.08\,\text{ms}^{-1}$, $\text{MAE} = 0.10\,\text{ms}^{-1}$) in the zonal and meridional directions, respectively. Both values fall within the

theoretically predicted range of $1 - 2\%$ for objects with a density of approximately $0.7\rho$, where $\rho$ is the seawater density (Wagner et al., 2022).

Focusing on the scale of each of the resulting terms in this linear regression, we observe a contrast between the predominant dynamics in the zonal and meridional direction. The mean zonal surface ocean current speed, $\langle\,|\,U_o\,|\,\rangle = 0.3\,\mathrm{ms}^{-1}$, is higher than the resulting mean wind contribution $\langle\gamma^x\,|\,U_w\,|\rangle = 0.07\,\mathrm{ms}^{-1}$, and at least one order of magnitude larger than the mean zonal Stokes drift speed ($\langle\,|\,U_s\,|\,\rangle = 0.04\,\mathrm{ms}^{-1}$). In the meridional direction, however, mean surface ocean currents speed, $\langle\,|\,V_o\,|\,\rangle = 0.08\,\mathrm{ms}^{-1}$, and wind effects $\langle\gamma^y\,|\,V_w\,|\rangle = 0.06\,\mathrm{ms}^{-1}$ are comparable, while the mean Stokes drift speed ($\langle\,|\,V_s\,|\,\rangle = 0.03\,\mathrm{ms}^{-1}$) is still smaller.

The random forest and support vector regression models of the total drifter velocity exhibit similar $R^2$, RMSE, and MAE from cross-validation, performing a better fit of the zonal drifter velocity than the meridional velocity (see the legend in Figure 2 and Figures SS1 and SS2 in the Supplementary material). Model-agnostic interpretation methods consistently identify ocean currents and wind as the main drivers of the drifter's motion, approximating its total velocity as a linear combination of these forces. The permutation feature importance plots of the machine learning models (Fig. 2) show there is a prominent signature of the tidal current in the zonal direction for the prediction of the drifter zonal velocity $U_d$, represented by the high RMSE increase of $U_o^{HP}$ ($0.37\,\mathrm{ms}^{-1}$ in random forest model, $0.93\,\mathrm{ms}^{-1}$ in support vector regression), followed by a smaller contribution from the zonal wind $U_{10}$ ($0.14\,\mathrm{ms}^{-1}$, $0.23\,\mathrm{ms}^{-1}$ respectively). The roles of these two variables are reversed for the prediction of the meridional drifter velocity $V_{10}$ ($0.15\,\mathrm{ms}^{-1}$, $0.43\,\mathrm{ms}^{-1}$) compared to the contribution from the meridional high-pass ocean currents $V_o^{HP}$ ($0.05\,\mathrm{ms}^{-1}$, $0.13\,\mathrm{ms}^{-1}$).

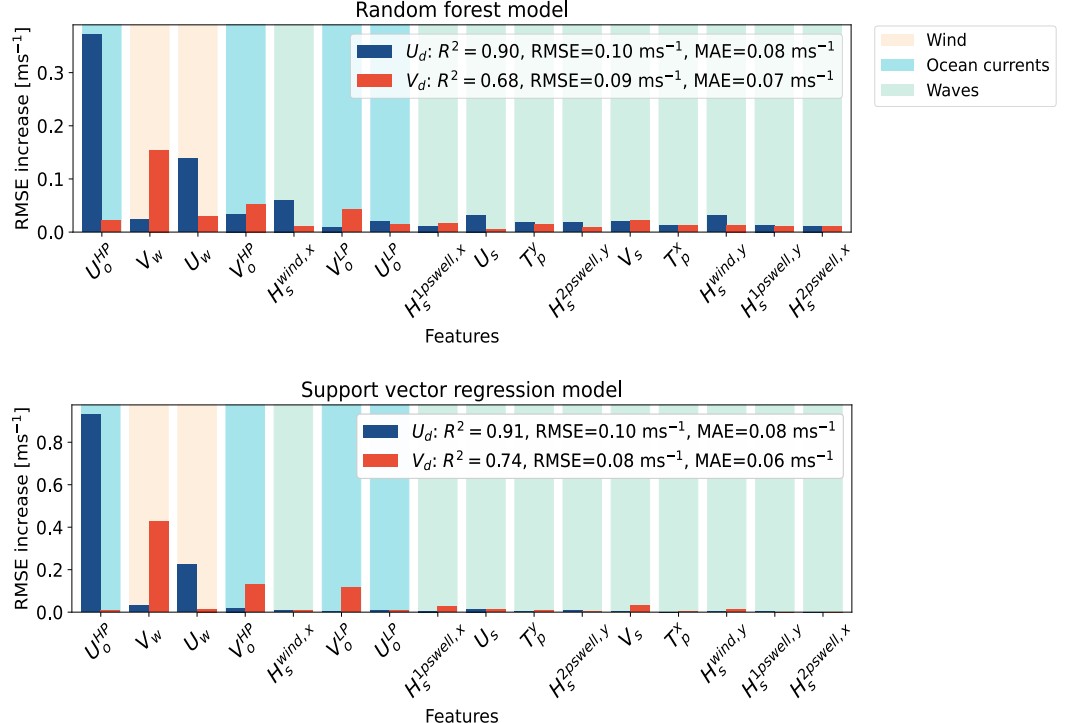

**Figure 2.** Permutation feature importance (RMSE increase) for a random forest and support vector regression models predicting zonal ($U_d$, blue) and meridional ($V_d$, red) total drifter velocity components calculated using central difference scheme. Bars represent the mean RMSE increase over 10 random permutations of each feature, which are ordered by decreasing importance. Error bars are omitted, as they account for less than 1% of the RMSE increase. A larger RMSE increase indicates greater feature importance, as shuffling a feature significantly worsens the model's prediction. Note the differences in the scale of the y-axis across the models. Features are shaded by type: ocean currents (blue), wind (beige), and waves (light green). Cross-validated metrics (coefficient of determination $R^2$, RMSE, and MAE) are shown in the legend.

ALE plots of the random forest model reveal the dependence between the drifter velocity components and the most influential features from the permutation feature importance plots: high-pass ocean currents and wind. We observe a linear relationship between the high-pass ocean currents and the parallel total drifter velocity component (Fig. 3a). Yet, we also observe small nonlinearities in the ALE plot of the meridional high-pass ocean currents $V_o^{HP}$ around zero and at the extremes of the feature
distribution, where uncertainty increases due to sparse data.

The ALE plots for the zonal wind velocity $U_{10}$ show that the effect on the zonal total drifter velocity $U_d$ becomes constant at extremes of the distribution, resembling a sigmoid function (Fig. 3b). This indicates that higher values of the zonal wind might not contribute to the same extent to the variance of the zonal total drifter velocity. The meridional wind $V_{10}$ exhibits a similar but weaker saturation effect on $V_d$ for strong westward winds ($< -7\,\mathrm{ms}^{-1}$), though data density is low in this regime. This
could be related to the fact that the friction velocity does not scale linearly with wind speed at high values due to increasing

surface roughness (Foreman and Emeis, 2010). However, experiments indicate that this nonlinearity typically occurs at wind speeds above the maximum observed in our data, which is $13.3\,\mathrm{ms}^{-1}$. Another possible explanation for the decoupling between winds and drifter velocity could be the transfer of kinetic energy from wind to the ocean, generating wind-driven currents. Yet this would require a high positive correlation between high wind speeds and surface currents, absent in our data (Fig. SS7 in

the Supplementary material). A further consideration is the difference in spatial resolution between the atmospheric ($0.25°$) and hydrodynamic ($0.027°$) models. While this discrepancy might seem relevant, the Rossby radius of the ocean remains much smaller than the synoptic scale, meaning the resolution difference likely has little effect on drifter velocity parameterisation.

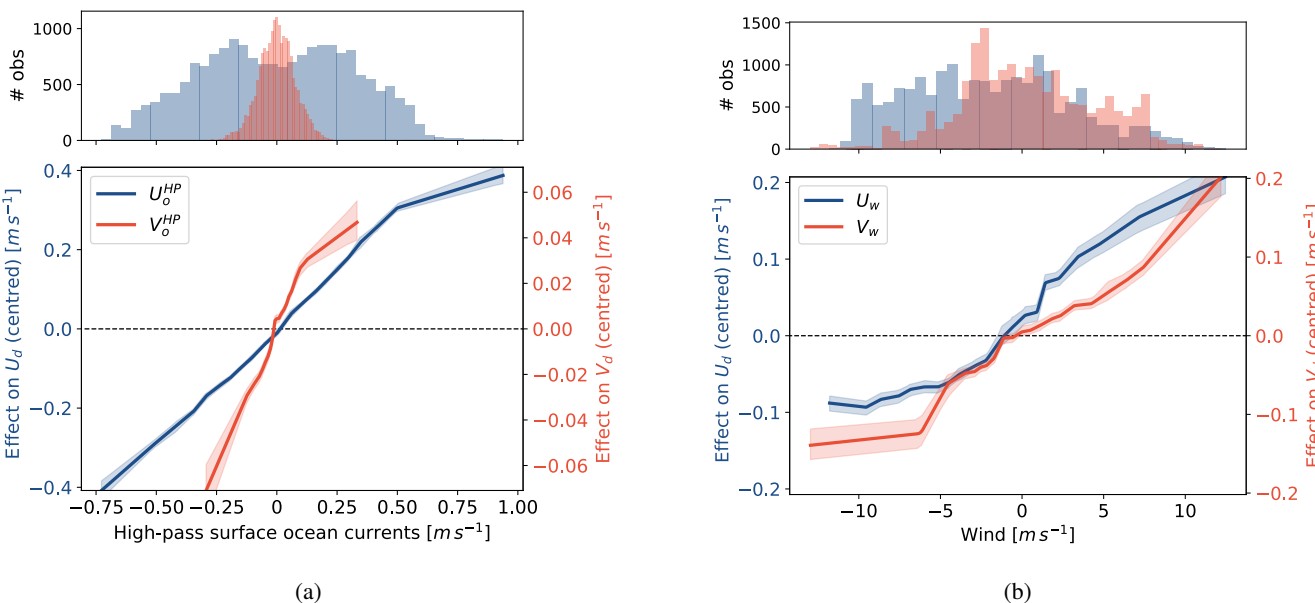

(a)  (b)

**Figure 3.** Accumulated Local Effect (ALE) plots for the zonal (blue, left y-axis) and meridional (red, right y-axis) total drifter velocity, calculated using central difference scheme, as a function of the corresponding parallel components of (a) the high-pass ocean currents and (b) the wind modelled by a random forest model. The shaded area represents the $95\%$ CI across 100 bootstrapped samples. The top histograms show feature distributions of the data. Negative velocity values for the zonal component represent westward motion, while negative meridional velocities represent southward motion.

### 4.1.2 Residual drifter velocity

By calculating the drifter residual velocity using Eq. (2), we effectively filter out dominant high-frequency ocean surface
currents (i.e., those shorter than the $M_2$ and $S_2$ tidal frequencies), allowing for a more focused analysis of the underlying net transport mechanisms. As seen in Fig. 4, the random forest and support vector regression models in that case assign the highest importance to the parallel components of the wind, Stokes drift velocity, and low-pass filtered ocean surface currents.

In the meridional direction, the wind $V_{10}$ (RMSE increase of $0.10\,\mathrm{ms}^{-1}$ in random forest, $0.45\,\mathrm{ms}^{-1}$ in support vector regression) and the low-pass current $V_o^{LP}$ ($0.07\,\mathrm{ms}^{-1}$, $0.30\,\mathrm{ms}^{-1}$ respectively) are the most important features in both models

for the drifter residual meridional velocity $\tilde{V}_d$. Yet, the contribution from the meridional Stokes drift $V_s$ is only significant in the random forest model compared to the support vector regression ($0.05\,\mathrm{ms}^{-1}$, $0.01\,\mathrm{ms}^{-1}$). This disagreement could stem from the definition of the permutation feature importance. A feature can only imply a meaningful importance if it is not strongly correlated with other features that also influence the target (Ewald et al., 2024). We find that meridional Stokes drift $V_s$ and meridional wind $V_{10}$ are highly correlated (Spearman correlation coefficient $\rho = 0.92$ (Spearman, 1904); see Fig. SS7 in the Supplementary material). The reason this is only captured by the random forest model and not the support vector regression is that in random forest models, correlated features can also replace each other if randomly left out by $m_{\mathrm{try}}$ for splitting. Instead, support vector regression performs a general transformation of feature space, where correlated features go into the same dimension and hence are less sensitive to data points further away from the general distribution.

In the zonal direction, the wind $U_{10}$ ( RMSE increase of $0.11\,\mathrm{ms}^{-1}$, $0.53\,\mathrm{ms}^{-1}$) and the Stokes drift $U_s$ ($0.06\,\mathrm{ms}^{-1}$, $0.18\,\mathrm{ms}^{-1}$) are the most important features for the zonal drifter residual velocity $\tilde{U}_d$ models. The contribution of the low-pass ocean currents $U_o^{LP}$ is also comparable ($0.06\,\mathrm{ms}^{-1}$, $0.11\,\mathrm{ms}^{-1}$).

From both the residual zonal and meridional drifter velocity models, we also find high permutation feature importance of variables that are not aligned with the velocity component, such as $V_{10}$ for $\tilde{U}_d$ ( RMSE increase of $0.02\,\mathrm{ms}^{-1}$, $0.09\,\mathrm{ms}^{-1}$) and $H_s^{\text{1st swell,x}}$ for $\tilde{V}_d$ ($0.09\,\mathrm{ms}^{-1}$ in the support vector regression model).

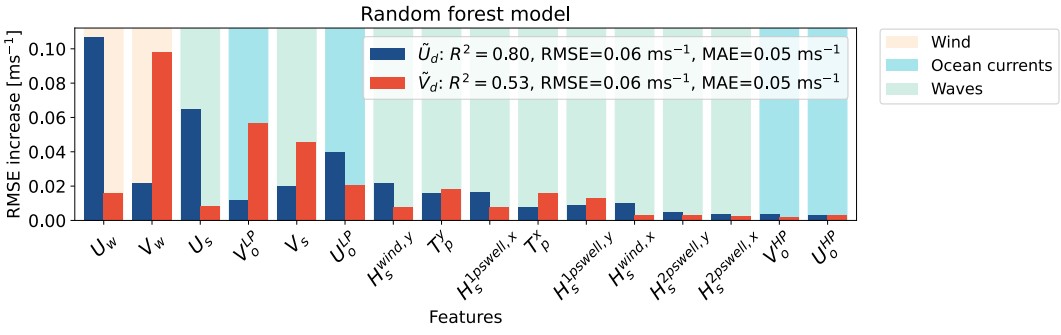

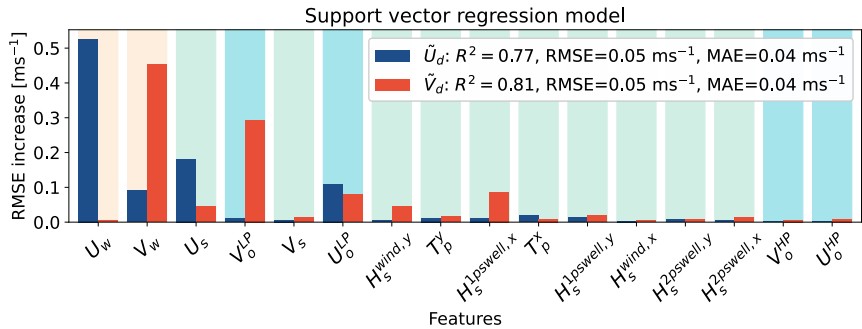

**Figure 4.** As in Fig. 2 but for the random forest and support vector regression models predicting the zonal ($\tilde{U}_d$, blue) and meridional ($\tilde{V}_d$, red) residual drifter velocity components.

Residual drifter velocity ALE plots show analogous dependence to the wind speed as the total drifter velocity: a linear regime for low speeds but plateauing at higher speeds (Fig. SS6 in the Supplementary material). Low-pass currents likewise show a linear relationship with the parallel residual velocity components where data density is high for speeds $< 0.10\,\mathrm{ms}^{-1}$ and saturation effects at the extremes of the distribution (Fig. 5a). Furthermore, we find that Stokes drift influences residual velocity noticeably above $0.05\,\mathrm{ms}^{-1}$ (Fig. 5b). Below this velocity threshold, the residual velocity remains largely unaffected, and the Stokes drift contribution to surface drifter transport is minimal. This suggests that in such a regime, the otherwise small influence of swell-induced Stokes drift may become relatively more significant and should not be neglected. At higher Stokes drift values, the drifter residual velocity increases approximately linearly in the zonal direction. A similar trend is observed in the meridional component, although with greater uncertainty due to the limited number of observations at high Stokes drift speed.

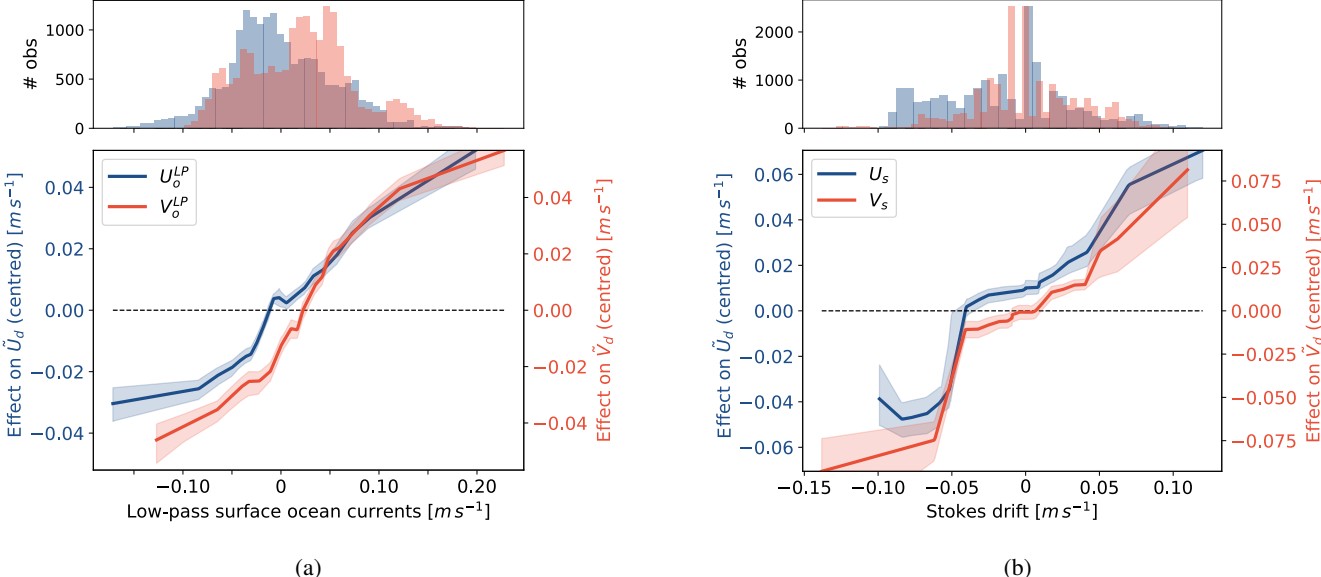

(a)                                                    (b)

**Figure 5.** Accumulated Local Effect (ALE) plots for the zonal (blue, left y-axis) and meridional (red, right y-axis) residual drifter velocity as a function of the corresponding parallel components of the (a) low-pass ocean currents and (b) Stokes drift modelled by a random forest model. The shaded area represents the 95% CI across 100 bootstrapped samples. The top histograms show feature distributions of the data. Negative velocity values for the zonal component represent westward motion, while negative meridional velocities represent southward motion.

## 4.2 Prediction of drifter trajectories

To evaluate the predictive skill of each model, we apply a leave-one-drifter-out strategy (see Sect. 3.3). The models are evaluated over a fixed prediction period of 60 days, after which the linear regression model typically predicts beaching. Repeating this process for each drifter yields 12 simulated trajectories per model. All the resulting trajectories succeed in reproducing tidal oscillations along the trajectory and large-scale patterns (e.g. the loop near 4°E longitude in Fig. 6).

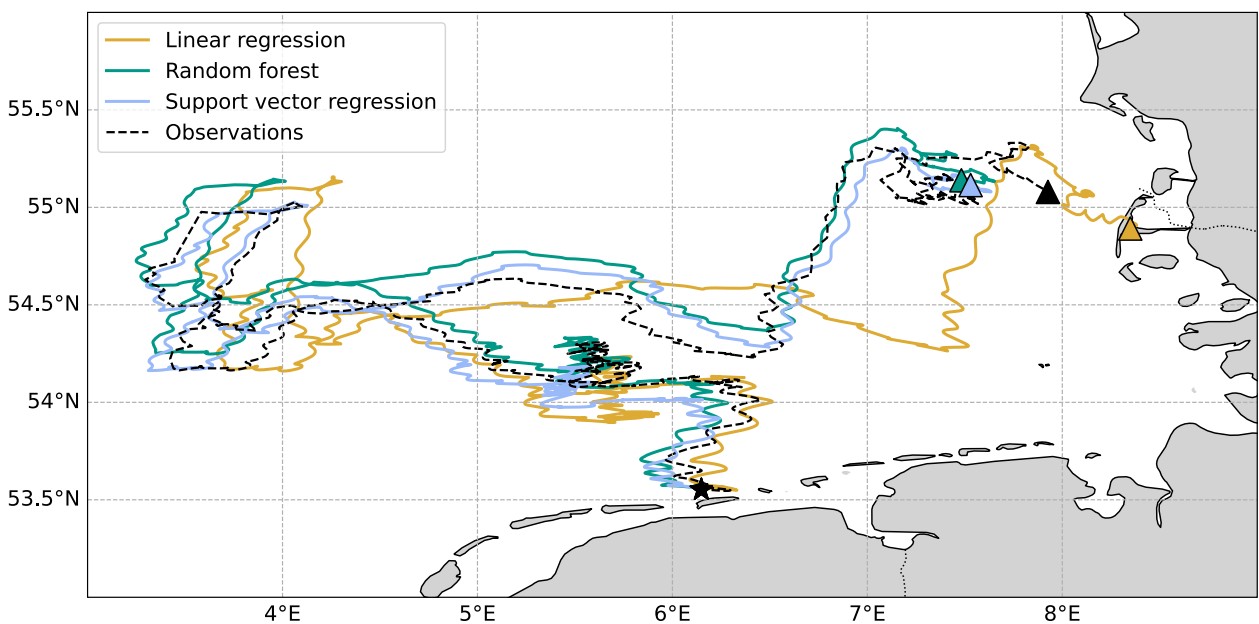

**Figure 6.** Example of a reconstructed drifter trajectory of over 60 days using linear regression (yellow), random forest (green) and support vector regression (blue) models using an integration timestep of dt $= 60$ s. The true measured drifter trajectory is shown in black. The wind slip coefficients used for this linear regression model are $\gamma^x = 1.34\%$ and $\gamma^y = 1.64\%$ in the zonal and meridional direction, respectively.

Comparing the mean cumulative distance of reconstructed trajectories from drifter velocity predictions in Fig. 7 demonstrates the advantages of using machine learning algorithms for prediction over linear regression for long-term predictions. Random forest achieves the lowest deviation from observations with $D = 10.8$ km and an interquartile range of all distances (IQR) of $3.2$ km, indicating superior accuracy and consistency across the test data (Fig. 7). The support vector regression model shows a lower performance, with a median cumulative separation distance of $D = 14.1$ km (IQR$= 2.3$ km), potentially due to the

fact that this model does not model high-order interactions by sub-partitioning the dataset into small sections like the random forest does. However, there is a risk of over-adaptation of the random forest model to the current training data; hence, the advantage of comparing the results from both models. Meanwhile, the linear regression model has the poorest performance with a median of $D = 27.2$ km and a higher IQR of $5.6$ km, indicating it is highly sensitive to the training data. The comparison of model performances using the Liu-Weisberg skill score also captures the superior performance of the machine learning

models compared to the linear regression model, assigning a skill score of $0.64 \pm 0.10$ to the random forest model, $0.55 \pm 0.09$ to the support vector regression model, and $0.10 \pm 0.16$ to the linear regression (Fig. SS8 in the Supplementary material). Deviations between the predicted and observed trajectories likely arise from biases in the training data caused by strong spatial correlations. As a result, (minor) velocity prediction errors can accumulate during integration, causing the simulated trajectories to diverge into regions not adequately represented in the training set.

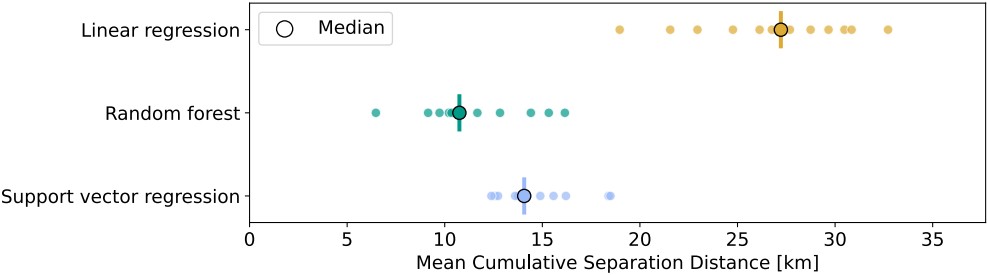

**Figure 7.** Mean cumulative separation distances using linear regression (yellow), random forest (green), and support vector regression (blue). Each model is trained on 11 trajectories and tested on the remaining one, with the process iterated so that each drifter is used once as the test set (leave-one-drifter-out cross-validation). Individual results for each drifter are shown as scatter points along with a median indicator.

The time evolution analysis of the differences between the observed and modelled trajectories from each of the models reveals that linear regression outperforms machine learning models for time scales smaller than 4 days (Fig. 8). After that onset time, the cumulative separation distance with respect to the observations increases over time for the linear regression predictions, while the error from the machine learning models remains constant, yielding a lower cumulative separation distance considering the entire trajectory. In the reconstructed trajectories of the drifters (e.g. Fig. 6), we observe that the linear regression model overestimates the zonal displacement, which could be caused by the fact that this model does not account for the decrease in the zonal wind contribution for higher wind speeds seen in the random forest ALE plots (Fig. 3b).

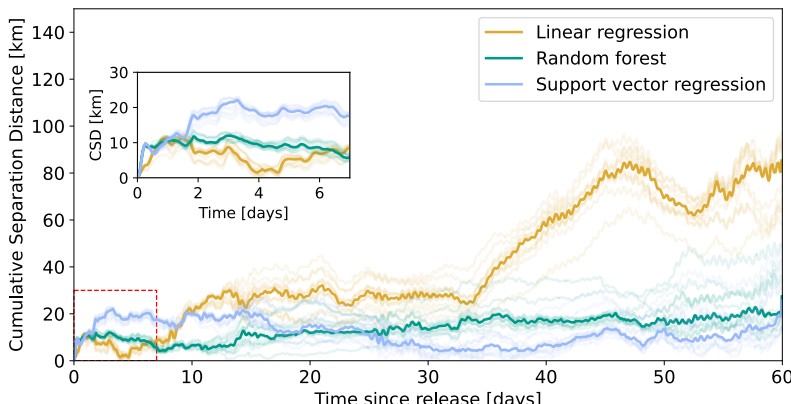

**Figure 8.** Time series of the cumulative separation distances using linear regression (yellow), random forest (green), and support vector regression (blue). Each model is trained on 11 trajectories and tested on the remaining one, with the process iterated so that each drifter is used once as the test set (leave-one-drifter-out cross-validation). Individual results for each drifter are shown as shaded lines, along with the median time series across drifters shown as solid lines. The small panel shows a close-up view for the first week since release.

Additionally, we apply the drifter trajectory model and evaluation procedure described in Sect. 3.3 to assess the effect of incorporating non-dynamical variables. Among the tested features, only the inclusion of the depth of the water column yields

an improvement in velocity prediction (Fig. 9). As expected, adding spatial coordinates such as latitude and longitude does not enhance model performance because the absolute location is not a relevant property, as it does not carry any physical meaning in a non-stationary flow. The parameterisation of wave-surfing transport via the Flipping Index results in a model with the same median predictive skill across drifter samples as the original random forest, but with somewhat reduced variance in prediction error. The random forest model trained to predict this index from hydrodynamic and atmospheric variables reveals a high permutation importance for wind speed and Stokes drift (Fig.E3, Fig.E4), highlighting their dominant role in the mechanisms associated with flipping events and, by extension, wave-driven transport.

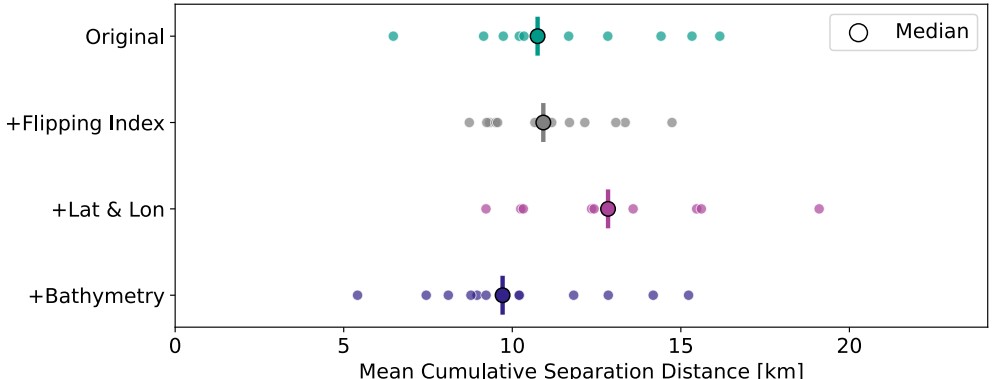

**Figure 9.** Mean cumulative separation distance metric applied to modelled trajectories from random forest models of the total drifter velocity with different sets of features: original model (green), including Flipping Index (grey), including latitude and longitude (magenta), and including bathymetry (dark purple). Each model is trained on 11 trajectories and tested on the remaining one, with the process iterated so that each drifter is used once as the test set (leave-one-drifter-out cross-validation). Individual results for each drifter are shown as scatter points along with a median indicator.

### 4.2.1 Linear models with alternative structure

Furthermore, we explore how to improve the predictive performance of the linear regression of the total drifter velocity using insights from the machine learning models. From the results of the ALE plots, the zonal and meridional drifter velocity in our data shows a sigmoid-like shape as a function of the zonal wind. Hence, we test a total drifter velocity linear regression model where the contribution of each wind component is modelled by a sigmoid function of its velocity, so that

$$\boldsymbol{u}_d = \boldsymbol{u}_o + \boldsymbol{u}_s + g(\boldsymbol{u}_w) + \varepsilon \quad \text{for} \quad g(\zeta) = \frac{a}{1 + e^{-b(\zeta - \zeta_0)}}, \tag{10}$$

yielding $a = 0.27\,\mathrm{ms}^{-1}$, $b = 0.31\,\mathrm{ms}^{-1}$, and $\zeta_0 = 1.79\,\mathrm{ms}^{-1}$ ($R^2 = 0.28$, RMSE $= 0.11\,\mathrm{ms}^{-1}$, MAE $= 0.07\,\mathrm{ms}^{-1}$) as best-fit parameters in the zonal direction and $a = 1.82\,\mathrm{ms}^{-1}$, $b = 0.04\,\mathrm{ms}^{-1}$, and $\zeta_0 = 12.2\,\mathrm{ms}^{-1}$ ($R^2 = 0.40$, RMSE $= 0.08\,\mathrm{ms}^{-1}$, MAE $= 0.07\,\mathrm{ms}^{-1}$) in the meridional. These fits improve upon the original linear model by reducing its bias, nearly halving the MAE.

The alternative approach using the relative wind yields best-fit coefficients of $\gamma^x = 1.39\%$ ($R^2 = 0.28$, RMSE $= 0.11\,\mathrm{ms^{-1}}$, MAE $= 0.11\,\mathrm{ms^{-1}}$), and $\gamma^y = 1.66\%$ ($R^2 = 0.41$, RMSE $= 0.08\,\mathrm{ms^{-1}}$, MAE $= 0.10\,\mathrm{ms^{-1}}$), also showing a small improvement in the fix with respect to the original linear model.

Following the method previously described for reconstructing the trajectories of the drifters from different models, we compare the predictive accuracy of the three linear regression models. From the resulting mean cumulative separation distance across drifter samples, we find that the sigmoid function parametrisation improves the linear regression predictions with $D = 15.8\,\mathrm{km}$ (IQR$= 3.7\,\mathrm{km}$) (Fig. 10). The relative wind parametrisation also shows an improvement in the predictions with $D = 25.8\,\mathrm{km}$ (IQR$= 3.3\,\mathrm{km}$). However, the analysis of the cumulative separation distance over time reveals that these differences between sigmoid and linear functions of the wind only emerge beyond $24\,\mathrm{h}$ after release (Fig. SS10 in the Supplementary material).

These findings suggest that the linear parameterisation of the wind contribution to drifter velocities, while effective at shorter timescales, may oversimplify the underlying physics. Indeed, this linear form resembles the functional form derived by theoretical studies using the Maxey-Riley framework for buoyant spherical (Beron-Vera et al., 2019) and non-spherical (Wagner et al., 2022) particles. However, as noted by Bos et al. (2025), the particle Reynolds numbers for the surface drifters used in this study in the southern North Sea are above the Stokes drag regime. Therefore, it is important to consider that air and water have different viscosities when determining the functional form of the wind contribution to drifter velocity, which may no longer be linear.

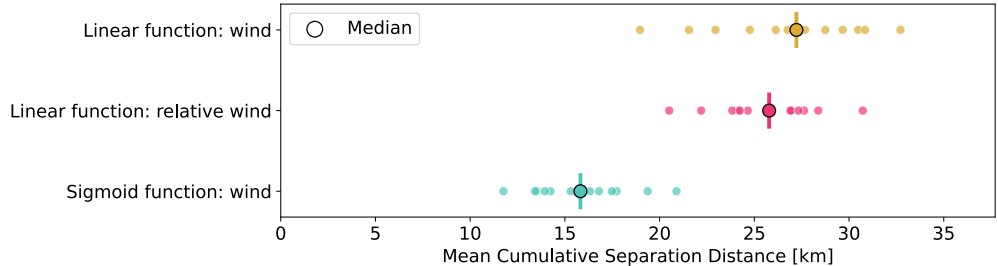

**Figure 10.** Mean cumulative separation distance metric applied to modelled trajectories using linear regression models with the original wind formulation (yellow) as per Eq. (3), the relative wind parametrisation (fuchsia) using Eq. (4), and the alternative sigmoid function of the wind (teal) from Eq. (10). Each model is trained on 11 trajectories and tested on the remaining one, with the process iterated so that each drifter is used once as the test set (leave-one-drifter-out cross-validation). Individual results for each drifter are shown as scatter points along with a median indicator.

### 4.2.2 Generalisability of the prediction results

We confirmed above, using a leave-one-drifter-out validation strategy, that machine learning models outperform linear equations predicting drifters in the North Sea. To test the transferability of those models into other regions, we use an additional data set from a drifter campaign in the Tyrrhenian Sea that started on 26 June 2025. Six surface drifters from MetOcean (2020) were deployed off the coast of Napoli. One stopped transmitting after one day, while the remaining five drifted for at least 40 days.

Full trajectories of these drifters are provided in Appendix F. We apply the North-Sea-trained models to predict the Tyrrhenian Sea drifter trajectories using the method and evaluation metrics described in Sect. 3.3. As predictor data, we use data from a coupled ocean—waves model of the Mediterranean Sea (Clementi et al., 2023) and the same atmospheric model described in Sect. 2.2. We also test the generalisability of the leeway method by applying the same windage coefficients derived from the North Sea data.

Trajectories were reconstructed over $48\,h$, with predictions iteratively repeated along the full drifter trajectory to obtain 20 predictions per drifter. The choice of a $48\,h$ prediction window is motivated by the aim of comparing its performance with that reported in other studies with different methodologies: Dagestad and Röhrs (2019) reported predicts CODE and iSphere drifter trajectories in the Norwegian coast using a physics-based linear model with different hydrodynamic models and Stokes drift configurations, while Grossi et al. (2025) builds an artificial neural network for CODE drifter trajectory predictions in the Gulf of Mexico based solely on previous latitude and longitude data.

**Table 2.** Evaluation metrics of the predictive performance of different models reproducing 5 surface drifter trajectories in the Tyrrhenian Sea: mean cumulative separation distance (D), RMSE error between observations and predictions, separation distance between observation and prediction after $48\,h$, and skill score. Machine learning models from this work are trained on the North Sea drifter dataset, and the windage coefficient for the linear regression is tuned for this data. The reported errors are the interquartile range (IQR)

| Studies | Models | Metrics | | | |
|---|---|---|---|---|---|
| | | D [km] | RMSE error [km] | Separation distance [km] | Skill score |
| This work | Linear regression | $10.5 \pm 0.3$ | $12.0 \pm 0.8$ | $19.1 \pm 0.8$ | $0.60 \pm 0.02$ |
| | Random forest | $\mathbf{9.6 \pm 0.1}$ | $\mathbf{11.0 \pm 0.1}$ | $\mathbf{17.4 \pm 0.1}$ | $\mathbf{0.66 \pm 0.01}$ |
| | Support vector regression | $9.7 \pm 0.3$ | $11.2 \pm 0.9$ | $18.9 \pm 0.9$ | $0.65 \pm 0.01$ |
| Dagestad and Röhrs (2019) | CODE | - | - | $22 - 28$ | $0.05 - 0.40$ |
| | iSphere | - | - | $20 - 27$ | $0.45 - 0.58$ |
| Grossi et al. (2025) | | - | $15 - 40$ | - | - |

Table 2 presents a comparison of the evaluation metrics reported in the literature with those obtained in the current study. Among the models tested, the random forest achieves the highest predictive performance, followed closely by the support vector regression, while the linear regression performs slightly less well. Overall, all models demonstrate a predictive skill after $48$, h consistent with previous studies. Figure G1 shows the predicted trajectory of a single drifter in the Tyrrhenian Sea to illustrate the results.

These findings indicate that the machine learning models trained on the North Sea data can largely reproduce the dynamics in the Tyrrhenian Sea despite the differences in the ocean dynamics between the two regions. This suggests a low degree of overfitting and over-adaptation to the specific conditions of the data of the campaign from the North Sea. The North Sea is a tidally dominated region, while the Tyrrhenian Sea is part of the Mediterranean Sea, a semi-enclosed basin with a thermohaline and wind-driven circulation with eddies as regular features (Rinaldi et al., 2010; Buffett et al., 2017).

## 5 Conclusions

This study analyses the effect of near-surface ocean currents, wave-induced motions, and wind drag on the trajectories of ultra-thin surface drifters to gain insight into the transport of buoyant objects at the ocean surface. We follow a data-driven approach and regress drifter velocity against instantaneous hydrodynamic and atmospheric conditions, and compare the established linear leeway model with two fundamentally different machine learning algorithms in terms of both inference of predominant forcing mechanisms and trajectory prediction performance.

At first order, machine learning models indicate that wind and ocean currents are linearly related to drifter velocity and are the most important features explaining the variability in velocity. These findings align with previous observational studies using leeway formulations (Breivik et al., 2011; Dominicis et al., 2016) and theoretical predictions from the Maxey-Riley framework (Beron-Vera et al., 2019). Stokes drift also contributes notably for low values of the residual (non-tidal) drifter velocity.

Non-linear behaviour emerges under strong wind, as revealed by the ALE plots of the drifter velocity (Fig. 3b). Incorporating this insight, we improve trajectory predictions by using a quasi-linear model with a non-linear wind term. While linear approaches remain advantageous in operational oceanography due to their simplicity and computational efficiency, we propose a hybrid framework that utilises interpretable machine learning methods to reveal functional relationships between drifter velocity and environmental forcing. These insights can guide the formulation of more accurate linear parameterisations.

When evaluating trajectory predictions from integrated modelled zonal and meridional velocities, the linear model performs reasonably well for predictions of less than 4 days but accumulates bias over longer periods. In contrast, machine learning models, especially the random forest, consistently outperform the linear baseline (Fig. 8). This suggests that more complex models might be needed to extend the forecasting horizon.

Feature engineering analysis shows that incorporating additional physically relevant information, such as water depth or a parameterisation of wave-surfing effects, further improves random forest performance. Meanwhile, adding non-physical features, such as longitude and latitude, degrades predictions. Finally, we test the generalisability of the models to a region with markedly different ocean dynamics, the Tyrrhenian Sea. Performing 48h-reconstructions of surface drifter trajectories demonstrates a predictive skill comparable to other state-of-the-art studies, indicating low overfitting to the North Sea training data and reinforcing the physical conclusions of the near-surface ocean dynamics of this study.

*Code availability.* Code used to conduct the experiment is available at https://github.com/jimena-medinarubio/ML_surface-drifters.git

*Data availability.* Drifter data in the North Sea is available at: https://doi.org/10.5281/zenodo.14198921. Drifter data in the Tyrrhenian Sea is available at: https://doi.org/10.5281/zenodo.17293098

## Appendix A:  Estimation of spatial coordinate errors

We estimate the uncertainty in the spatial coordinates reported by the GPS system in the drifters from the errors in distribution of measurements during an experiment. We position Stokes drifters, identical to the ones used in this study, over a flat surface on land. A total of 84 coordinate data points were measured from each drifter, derived from two distinct measurement rounds. The first round comprised a 24-hour cycle with a 30-minute transmission frequency, and the second was a 3-hour cycle utilising a 5-minute frequency (Schneiter and van Sebille, 2023). For each drifter, we compute the deviation of the longitude and latitude measurements with respect to their mean during the combined measuring periods and approximate their density distribution to continuous using the Kernel Density Estimation method. The resulting curves can be observed in Fig. A1, which highlights the mean standard deviation of the measurement distribution across drifters: $8.4\,\mathrm{m}$ in the latitudinal direction, and $6.5\,\mathrm{m}$ in the longitudinal direction.

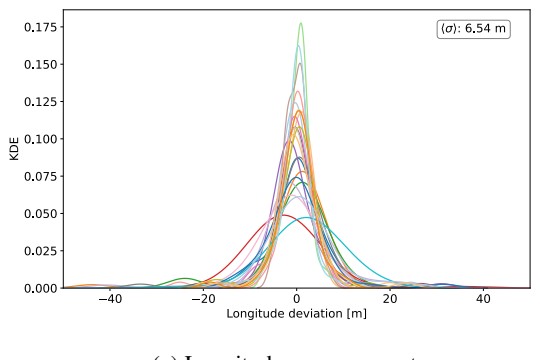
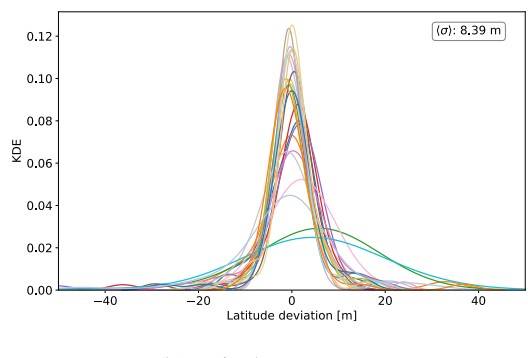

(a) Longitude measurements          (b) Latitude measurements

**Figure A1.** Distribution of the deviation of (a) longitude and (b) latitude measurements with respect to their mean from 24 colour-coded stationary surface drifters. Data were collected during a 24-hour cycle (30-minute frequency) and a 3-hour cycle (5-minute frequency). The distributions are estimated using Kernel Density Estimation (KDE). The mean standard error $\sigma$ across all drifter distributions is included in the legend.

## Appendix B:  Power spectral analysis of the drifters' velocity

We use power spectral analysis to identify the dominant tidal harmonic using two complementary techniques: Fast Fourier Transform (FFT) and Morlet Wavelet analysis. For FFT analysis, uniform time spacing of the measurements is required, so perform the analysis to two different periods of time independently: from day 6-26, when the sampling period is $30\,\mathrm{min}$, and from day 26 onwards, when the sampling period is $3\,\mathrm{h}$. We also performed a Morlet Wavelet spectral analysis to investigate temporal variations in the frequency spectrum, as the time series spans more than one spring-neap tidal cycle (Meyers et al., 1993). Unlike FFT, this approach does not require time resampling, allowing the detection of higher-frequency harmonics without compromising the integrity of the original temporal resolution. The analysis of the 5-minute period (from the beginning

of the time series to day 6) has not been included as the time interval between samples is more irregular (not exactly $300\,\mathrm{s}$, with an average standard deviation of $\pm 22\,\mathrm{s}$). This irregular sampling complicates the alignment and averaging of results across drifters. The corresponding text in the manuscript has been updated.

Apart from the predominant signals of the $M_2$ and $S_2$ tidal constituents, we also observe a weaker contribution from the high-frequency lunar tidal constituent $M_4$ in the Morlet Wavelet graph when the sampling period satisfies $\Delta t \leq 30\,\mathrm{min}$ (Fig. B1). This is due to the fact that for a sampling period of $3\,\mathrm{h}$, the Nyquist period is $6\,\mathrm{h}$, which closely matches the period of the $M_4$ signal ($6.2\,\mathrm{h}$). Hence, during the period when the sampling frequency is coarsest, the time resolution of the observations is barely enough to detect this signal.

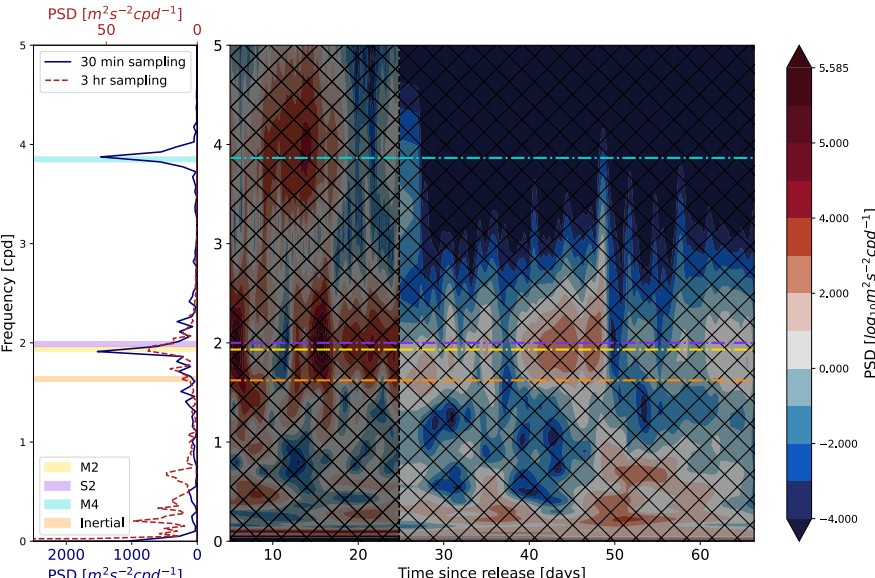

**Figure B1.** Average power spectrum across all drifters' speed using Fast Fourier Transform (left) and Morlet Wavelet (right). The FFT shows two spectra corresponding to the analysis over a time period with sampling frequency of $30\,\mathrm{min}$ (blue) and $3\,\mathrm{h}$ (red) with their respective x-axis. The Morlet Wavelet graph is a concatenation of the results for both periods, separated by a discontinued line The frequencies of the main tidal harmonics found in the German Bight region ($M_2$, $S_2$, $M_4$) are highlighted in colours (yellow, purple, and light blue respectively) as well as the inertial frequency at $54°$ latitude (orange).

## Appendix C: Spatiotemporal block cross-validation strategy

We use a spatiotemporal block cross-validation strategy to mitigate the impact of temporal autocorrelation and spatial correlation in the dataset (Wadoux and Heuvelink, 2023). Data is first aligned in time and then segmented into blocks. Subsequently, a standard k-fold cross-validation approach is applied by splitting the shuffled blocks into five folds. During each iteration, four folds are used for training, and the remaining fold is used for validation, ensuring all blocks are eventually tested.

The duration of each block corresponds to the average autocorrelation time of the target variable across all drifters using the e-folding scheme. The autocorrelation functions are found by calculating the Pearson correlation at time lags ranging from $1 - 100\,\mathrm{h}$ using *statsmodels* Python package (Skipper and Josef (2010), version 0.14.2). The resulting correlograms are shown in Fig. C1, and the resulting autocorrelation times are summarised in Table C1.

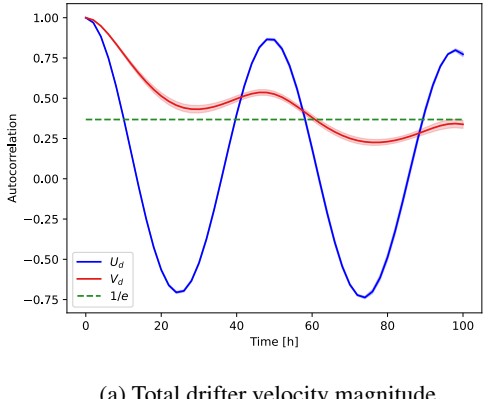
(a) Total drifter velocity magnitude

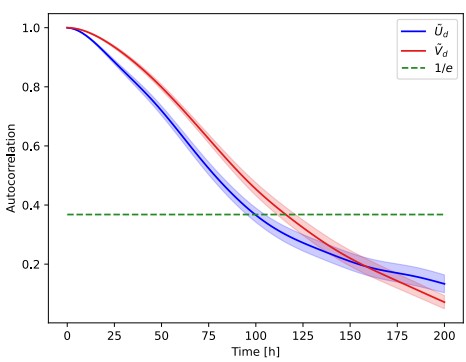
(b) Residual drifter velocity magnitude

**Figure C1.** Correlogram of the (a) total and (b) residual drifter velocity components across all drifters (zonal is shown in blue and meridional in red). The shaded region represents the standard deviation from the autocorrelation functions of each drifter velocity component.

**Table C1.** Autocorrelation times for different drifter velocity components as a result of using e-folding scheme.

| Velocity component | Autocorrelation time |
| --- | --- |
| $U_d$ | 12 h |
| $V_d$ | 62 h |
| $\tilde{U}_d$ | 101 h |
| $\tilde{V}_d$ | 117 h |

## Appendix D: Support vector regression hyperparameters

Support vector regression models use kernel functions to transform the data into a higher-dimensional space. In this higher-dimensional space, the support vector regression algorithm attempts to find a hyperplane that best fits the data, while allowing for some deviations from the actual observations, controlled by the parameter $\varepsilon$, called the margin of tolerance (Hastie et al., 2009). We choose the Radial Basis Function (RBF) kernel to build our models, which is commonly used for its ability to capture non-linear relationships between data points. The RBF kernel is defined as:

$$K(\mathbf{x}, \mathbf{x}') = e^{-\gamma \|\mathbf{x} - \mathbf{x}'\|^2}$$

where $\|\mathbf{x} - \mathbf{x}'\|^2$ is the squared Euclidean distance between two feature vectors $\mathbf{x}$ and $\mathbf{x}'$, and $\gamma$ is a parameter that controls how much influence a single training point has (Pedregosa et al., 2011). Additionally, the parameter $C$ alters the decision surface's smoothness that fits the target data in the hyperplane. The best fit from the SVR models of the total and residual velocity components, and Flipping Index, is found for the hyperparameters included in Table D1 using *GridSearchCV* functionality (Pedregosa et al., 2011).

**Table D1.** Value of the hyperparameters $C, \gamma, \varepsilon$ used in the Support Vector Regression models using a Radial Basis Function kernel to fit the total drifter zonal and meridional velocity ($U_d, V_d$), residual drifter zonal and meridional velocity ($\tilde{U}_d, \tilde{V}_d$).

| Target variable | C | $\gamma$ | $\varepsilon$ |
|:---:|:---:|:---:|:---:|
| $U_d$ | 0.30 | $2.0 \times 10^{-3}$ | 0.10 |
| $V_d$ | 0.30 | $1.0 \times 10^{-3}$ | $9.5 \times 10^{-2}$ |
| $\tilde{U}_d$ | 0.30 | $2.5 \times 10^{-3}$ | $9.0 \times 10^{-2}$ |
| $\tilde{V}_d$ | 0.70 | $3.5 \times 10^{-3}$ | 0.15 |

## Appendix E: Flipping Index model

We define a new metric named Flipping Index ($F$) to quantify the proportion of changes in the drifters' orientation or flips observed in subsequent measurements. To derive this index for each trajectory, the flips of the drifters are identified over time as the changes in the orientation signal, resulting in a binary variable $f(t)$ that equals 1 if a flip is observed at time $t$ and 0 otherwise. Then, these flips' time series are convolved with a sliding window of size $n(t)$ that increases with the sampling frequency of the measurements. Hence, the Flipping Index is defined as:

$$F(t, n(t)) = \sum_{i=-\frac{n(t)}{2}}^{\frac{n(t)}{2}} f(t+i), \quad n(t) = \frac{L}{\Delta t(t)} \tag{E1}$$

where $L$ is the fixed length of the temporal window, and $\Delta t(t)$ is the sampling frequency at time $t$, which increases along the drifters' trajectory. The Flipping Index is computed using a window size with $L = 3\,\mathrm{h}$, which corresponds to the highest sampling frequency of the drifter dataset. This choice was based on an analysis of window sizes ranging from $1 - 8\,\mathrm{h}$, which revealed only minor variations in the magnitude of the Flipping Index peaks (Fig. E1).

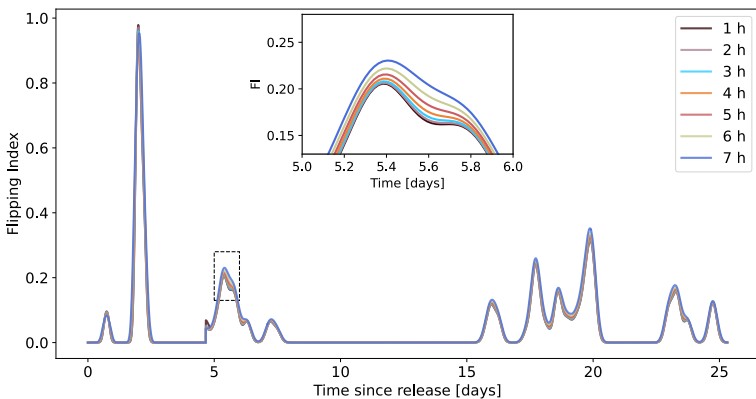

**Figure E1.** Time series of the Flipping Index of a single example surface drifter using different choices of temporal window size ranging from $1 - 8\,\text{h}$. Close-up view between day $5, 12\,\text{h}$ after release and day 8 shows small variations at the peak depending on the choice of the window size.

To ensure a continuous representation of the Flipping Index and reduce sensitivity to the non-uniform timestep, we apply a Gaussian smoothing filter. The standard deviation of the filter is set to $\sigma = \langle \delta t \rangle / 2$, where $\langle \delta t \rangle$ represents the mean time interval between consecutive flips. The index is subsequently normalised by the maximum number of flips observed across all drifters, resulting in a dimensionless measure ranging from 0 to 1. In this framework, a Flipping Index of 1 corresponds to 10 flips occurring within a window size with $L = 3\,\text{h}$. The Flipping Index is evaluated only for time periods with sampling frequency $\Delta t \leq 30\,\text{min}$. At lower sampling frequencies, the number of detected flips is likely strongly underestimated, and the temporal resolution becomes insufficient to capture submesoscale variability (Essink et al., 2022). Fig. E2 presents the resulting time series of the mean Flipping Index across all drifters. The relatively small standard deviation indicates that drifters tend to flip simultaneously, which can be attributed to their spatial proximity and shared exposure to similar hydrodynamic conditions throughout most of their trajectories.

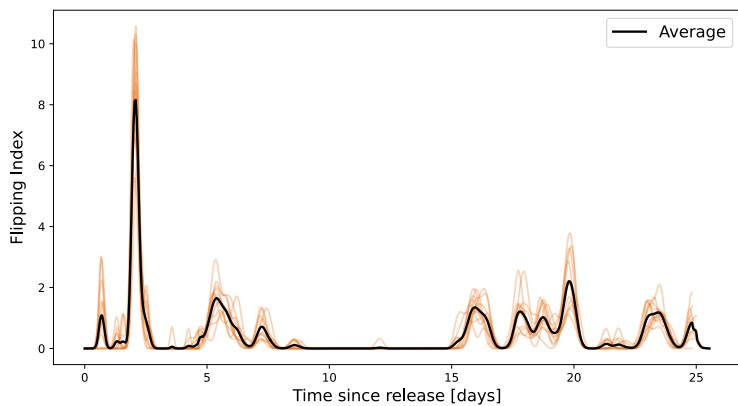

**Figure E2.** Time series of the Flipping Index from the 12 surface drifters (orange lines), along with the average across them (black line). The index is computed based on changes in drifter orientation relative to the ocean surface (i.e. flips) over a 26-day trajectory in the southern North Sea. Peaks indicate periods of increased flipping, likely caused by intense wave activity during storm conditions.

To assess whether including non-physical variables improves the random forest model's predictive skill, we create an additional model that fits the Flipping Index for a given hydrodynamic and atmospheric conditions, and then include this index as a feature in the prediction of the total drifter velocity, which parametrises stormy conditions. However, only $28\%$ of the data points yield a non-zero Flipping index (i.e., it is a zero-inflated variable), so the standard random forest algorithm would have difficulties predicting these zeroes (Fig. S4a). In order to solve this issue, we use a hurdle or two-step model that first creates a binary variable of the Flipping Index using a threshold, which we establish at $F = 0.05$, and trains a random forest classifier to predict instances when the Flipping Index is non-zero (Prasad et al., 2006). Then, we train a random forest regressor with the predicted non-zero Flipping Index data points to learn the relationship between the climate variables and the magnitude of the Flipping Index. The resulting permutation feature importance analysis reveals that most of the variance of the binary Flipping Index variable is explained by a combination of zonal wind, low-pass currents, and various wind-related variables (Fig. E3), while the highest importance for the model predicting the magnitude of this Flipping Index is assigned to the Stokes drift and the wind (Fig. E4).

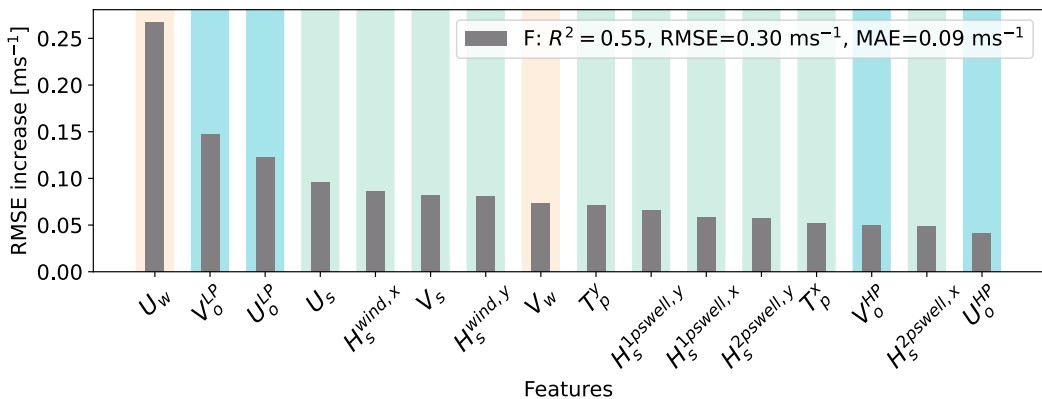

**Figure E3.** Permutation feature importance (RMSE increase) for a random forest classifier predicting the binary Flipping Index of the drifters in a hurdle model. Grey bars represent the mean RMSE increase over 10 random permutations of each feature, which are ordered by decreasing importance. Features are colour-coded by a shadow: blue indicates ocean current-related variables, beige corresponds to wind, and teal to wave parameters. Cross-validated model performance metrics (coefficient of determination $R^2$, RMSE, and MAE) for each velocity component are shown in the legend.

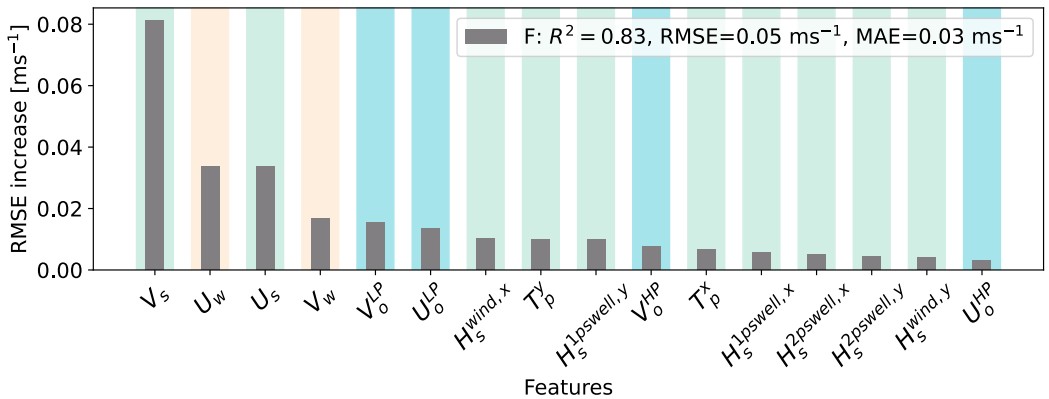

**Figure E4.** As in Fig. E3 for a random forest regressor predicting the magnitude of the Flipping Index of the drifters in a hurdle model for non-zero predictions from the classifier.

This variable is then used in the test case as a feature to train a random forest model to predict the trajectories of the drifters. For a new location of the predicted drifter, this trained Flipping Index model takes as input the interpolated hydrodynamic and atmospheric features and predicts the amount of flipping at that location. Then, this information is also included as input for the total drifter velocity model that predicts the velocity of the drifter used in the advection scheme.

## Appendix F: Drifter trajectories in the Tyrrhenian Sea

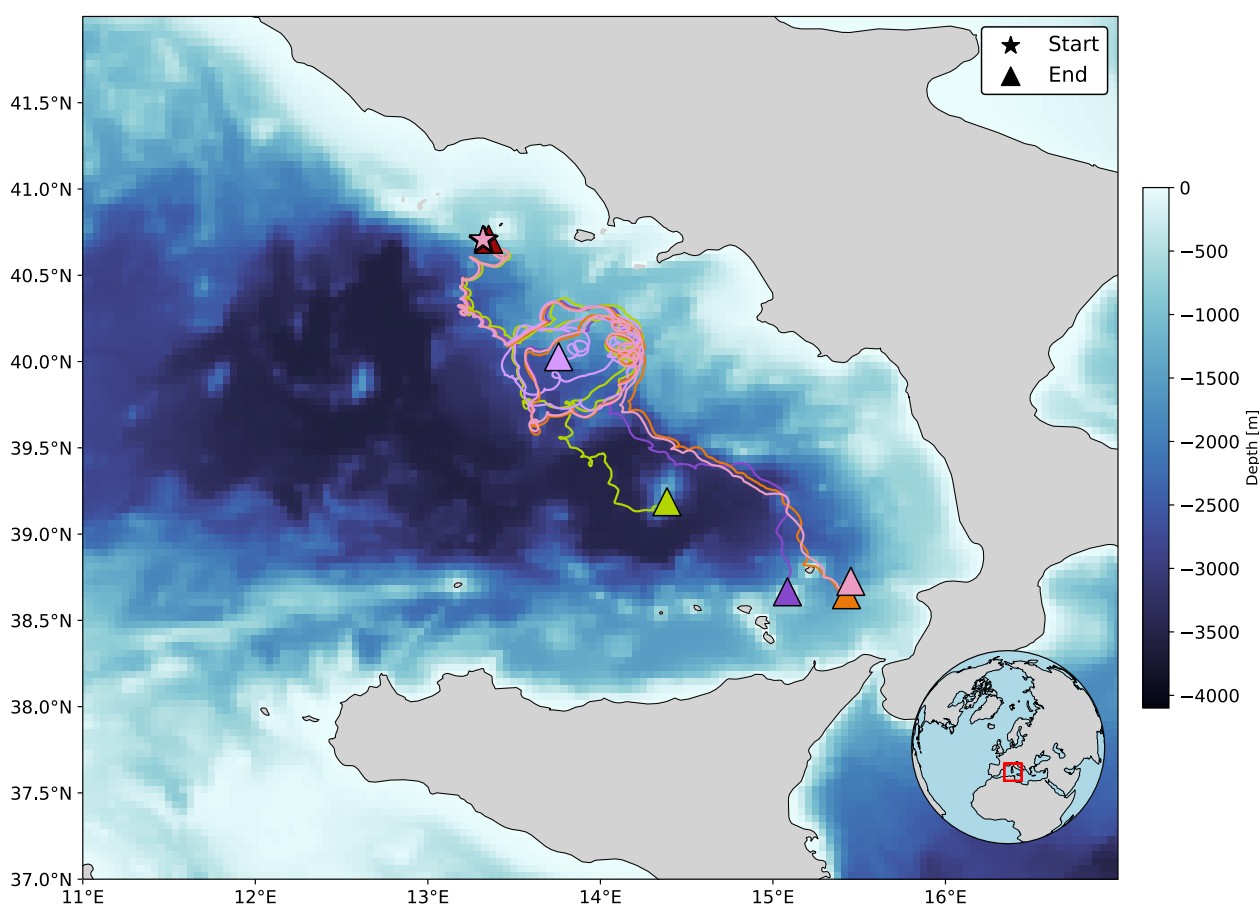

**Figure F1.** Trajectories over 49 days of 5 colour-coded surface drifters in the Tyrrhenian Sea. Drifters were deployed on 26 June 2025 in the Gulf of Naples in three different clusters spaced 1 km apart. Starting and ending positions are marked with stars and triangles, respectively. Background colourmap shows the bathymetry of the Tyrrhenian Sea from the Mediterranean Ocean Physics Analysis and Forecast model with a horizontal resolution of 0.042° (Clementi et al., 2023). The study site location in the Tyrrhenian Sea is highlighted by a red rectangle on the orthogonal projection of the Northern Hemisphere in the bottom right corner.

## Appendix G: Prediction results of trajectories in the Tyrrhenian Sea

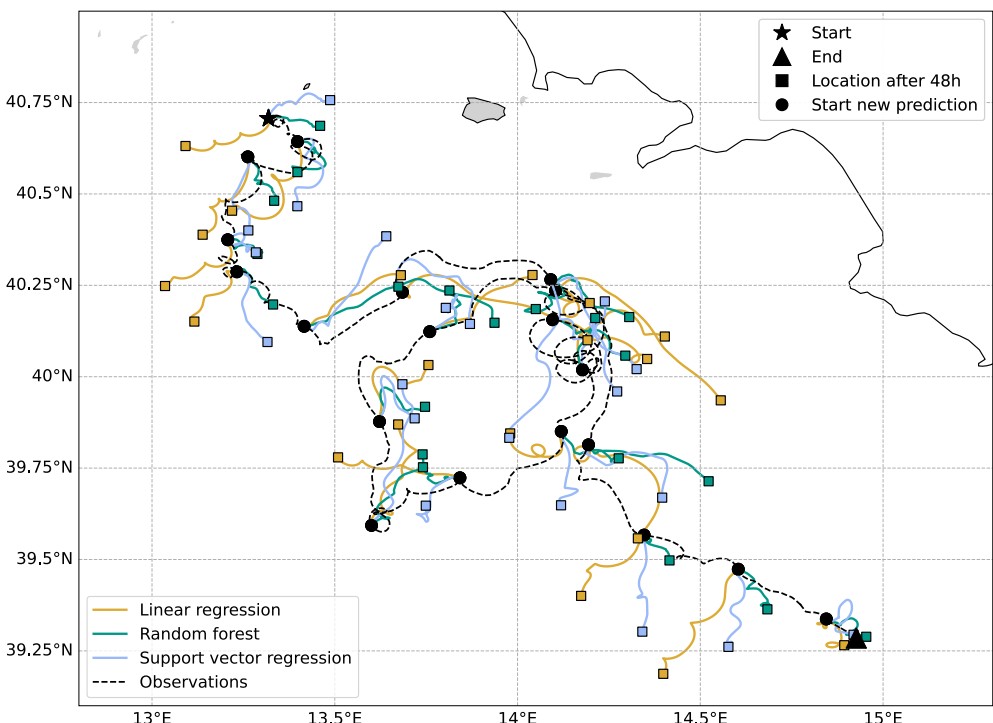

**Figure G1.** Example of an iterated reconstructed surface drifter trajectory with a forecasting window of $48\,\mathrm{h}$ using linear regression (yellow), random forest (green) and support vector regression (blue) models using an integration timestep of $\mathrm{dt} = 60\,\mathrm{s}$. The true measured drifter trajectory is shown in a black solid line. The wind slip coefficients used for this linear regression model are $\gamma^x = 1.34\%$ and $\gamma^y = 1.64\%$ in the zonal and meridional direction, respectively. The beginning of each new prediction period is marked by a solid black dot.

*Author contributions.* JMR designed and conducted the study, with steering and discussion from EvS, TvdB, and MN. All authors contributed to the manuscript.

*Competing interests.* At least one of the (co-)authors is a member of the editorial board of Ocean Science.

*Acknowledgements.* We would like to thank the team at IMAU who helped with the deployment of the drifters, including Bas Altena, Meike Bos, Michael Denes, Claudio Pierard, Daan Reijnders, Marc Schneiter, Jelle Soons, Margo van Asschenbergh, and Anna van Herwijnen. This publication is part of the project "*Tracing Marine Macroplastics by Unraveling the Ocean's Multiscale Transport Processes*" with file number VI.C.222.025 of the research programme Vici ENW which is (partly) financed by the Dutch Research Council (NWO). The Stokes drifters were purchased with support from the Paying-it-forward campaign of the Utrecht University Fund.

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
