# Peer review of "Using surface drifters to characterise near-surface ocean dynamics in the southern North Sea: a data-driven approach"

_EGUsphere, 2025_

## Author Comment (AC1)

**REVIEWER #1**

In this paper the authors use different machine learning models to characterize near surface ocean dynamic. The authors launched several undrogued surface drifters in the North Sea released from the coast of Netherlands, tracking their position with GNSS. Then, several variables (including variables derived from wind, oceanic currents and waves) from different research products are used as inputs in three machine learning models (linear regression, random forest and support vector machine) to predict drifter velocities. Permutation feature importance and ALE plots are then used to explain the importance of the input variables in predicting the drifter velocities.

The authors claim two different results in the conclusions.

The first one is the efficacy of the proposed analysis method. The use of techniques of explainable machine learning to investigate surface ocean dynamic is interesting and sufficiently novel. I have no objections for this part.

The second one is the accuracy of the proposed method in inferring drifter trajectories. This is, in my opinion, the weakest part of the paper. Albeit the numerical results support the conclusions of the authors, the trajectory dataset is very small, consisting of twelve drifters, released the same day at 250 meters of distance. As can be seen from the figures in the paper, the trajectories are higly correlated, meaning that the dataset lacks the variety needed to ensure sufficient generalization. In this condition the risk of overfitting a model during training is very high, and this problem is neither mentioned nor addressed in the paper.

The reason why the trajectory integrated using the linear model outputs is much more different from the other might be because, due to being a simpler model, it overfitted less than random forest and support vector regression.

I still think that integrating the trajectories using the model outputs is a reasonable benchmark, if the scope of the models is to explain the correlations between input variables and predicted drifter velocities.
In order to claim that the model is able to generalize beyond the twelve drifters presented in the paper, a test using some other drifter release (from some other starting position, in some other period) should be necessary.

I understand that drifter release is a demanding task, and I am obviously not asking the authors to plan further releases. However, in order to better understand the generalization limits, if other surface drifter trajectories are available to the authors, I suggest to test the trained models to reproduce them. If this is not possible, I expect that these concerns are better addressed in the conclusions.

At the very least, the model-integrated trajectories should be compared with trajectories simulated using the ocean velocities given as input to the machine learning models, using some classical integration scheme such as RK4 or RK45.

As a last note, even if the models are actually overfitting the data, this is not an issue for the first scope of the paper (predictor-velocity analysis), since the analysis is focused on this particular dataset and has no claim of generalization. Some degree of overfitting might even be considered beneficial.

We are grateful to the reviewer for highlighting this point and we agree that the implication that these models are rigorous as a standalone forecasting mechanism and could be suitable to predict drifter velocities in other regions beyond the southern North Sea has not been supported with evidence.

While the full dataset was not available at the time of submission, we have since obtained measurements from a drifter campaign with the same surface drifters recovered from the North Sea experiment. A research team from the Department of Marine Animal Conservation at the *Stazione Zoologica Anton Dohrn Napoli,* led by Dr. Hochscheid, deployed six drifters off the coast of Napoli on 26 June 2025 to compare the drifting behaviour of buoyant objects and turtles (van Sebille, 2025). One of the drifters stopped transmitting after one day, while the remaining five drifted for at least 40 days in the Tyrrhenian Sea.

In the revised manuscript, we have applied the North-Sea-trained models to predict these unseen trajectories using the method and evaluation metrics described in Section 3.3. We also test the generalisability of the leeway method by applying the same windage coefficients derived from the North Sea data. Trajectories were reconstructed over 48h with predictions iteratively repeated along the full drifter trajectory to obtain 20 predictions per drifter. The choice of a 48h prediction window is motivated by the aim of comparing its performance with that reported in other studies that aim to forecast on unseen data: (Dagestad & Röhrs, 2019) reported 20-40km separation distance after 48h for CODE and iSphere drifters in the Norwegian coast using a physics-based linear model, while (Grossi et al., 2025) found 20-50 km RMSE between prediction and observation for an artificial neural network model of the Gulf of Mexico.

The random forest model achieves the best performance (separation distance of 17.4 km, RMSE of 12 km, Mean Cumulative Separation Distance of 9.6 km), followed closely by the support vector regression model (18.9 km, 11.2 km, 9.7 km). The linear regression performs slightly worse (19.1 km, 12.0 km, 10.5 km). We also observed this good performance of linear regression for short-term prediction (Fig. 8) but found that errors rapidly propagate for timescales greater than one week. Nevertheless, compared with values from literature studies using both physics-based and machine learning models, all models show good forecasting skill after 48h.

These results indicate that the machine learning models trained on the North Sea data can reproduce the drifter behaviour in the Tyrrhenian Sea despite the differences in the ocean dynamics between the two. The North Sea is a tidally dominated region, while the Tyrrhenian Sea is part of the Mediterranean Sea, a semi-enclosed basin with a thermohaline and wind-driven circulation with eddies as regular features (Buffett et al., 2017).

All this additional analysis and results have been included in an additional section called "4.2.2. Generalisability of prediction results" in the tracked-changed manuscript. The conclusion has also been changed accordingly to include these new insights.

We also appreciate the reviewer's concern regarding potential overfitting. We would like to clarify that the cross-validation framework is specifically designed to detect and prevent overfitting by evaluating the model on data that is not used for training. We further strengthened this procedure by implementing a block cross-validation strategy, which ensures that training and validation is temporally independent. This prevents information leakage between folds and avoids the optimistic bias that can arise from autocorrelated data. We have changed lines 215-216 to clarify this:
*"(...) ensuring that any two points separated by more than this time range can be considered statistically independent and thus suitable for validation **This approach substantially reduces the risk of overfitting, as it prevents information leakage between training and validation sets and ensures that model performance is assessed on truly unseen, time-independent samples.** For a detailed explanation …."*

However, the reviewer is right to highlight that there might still be spatial correlation between the training data and the test with the drifters in the Tyrrhenian Sea addresses this concern.

**REFERENCES**

Buffett, G. G., Krahmann, G., Klaeschen, D., Schroeder, K., Sallarès, V., Papenberg, C., Ranero, C. R., & Zitellini, N. (2017). Seismic Oceanography in the

Tyrrhenian Sea: Thermohaline Staircases, Eddies, and Internal Waves. *Journal of Geophysical Research: Oceans*, *122*(11), 8503–8523.

https://doi.org/10.1002/2017JC012726

Dagestad, K.-F., & Röhrs, J. (2019). Prediction of ocean surface trajectories using satellite derived vs. Modeled ocean currents. *Remote Sensing of*

*Environment*, *223*, 130–142. https://doi.org/10.1016/j.rse.2019.01.001

Grossi, M. D., Jegelka, S., Lermusiaux, P. F. J., & Özgökmen, T. M. (2025). Surface drifter trajectory prediction in the Gulf of Mexico using neural networks.

*Ocean Modelling*, *196*, 102543. https://doi.org/10.1016/j.ocemod.2025.102543

van Sebille, E. (2025). *Tyrrhenian Sea drifter trajectories 2025*. Zenodo. https://doi.org/10.5281/zenodo.17293098

---

## Author Comment (AC2)

**REVIEWER #2**

This study investigates near-surface ocean dynamics in the southern North Sea using a data-driven approach based on the trajectories of 12 ultra-thin surface drifters deployed over a period of 68 days. The authors combine environmental data, such as wind, wave, and ocean current fields, with a linear leeway parameterisation and two machine learning models to predict drifter velocities. The study identifies the dominant physical drivers of drifter motion. It shows that zonal movement is largely driven by tidal currents, while meridional movement is more strongly influenced by wind. The machine learning models outperform linear models in long-term trajectory prediction, and their interpretability tools reveal important nonlinear effects, such as wind saturation. These insights are then used to improve the linear model by incorporating a nonlinear wind response. The study demonstrates how a data-driven framework, grounded in observational data and enhanced by machine learning, can deepen our understanding of surface transport mechanisms and improve predictive skill in operational oceanography.

The manuscript presents a well-designed and timely contribution that combines novel observational data with explainable machine learning techniques. The drifters used provide rare information on the very surface layer of the ocean, and the methodological approach is described in great detail and with clarity. Figures are clear and support the narrative well. A particular strength is the use of interpretability tools such as permutation feature importance and ALE plots, which link machine learning results to physical understanding. The study is highly relevant for both fundamental ocean dynamics and practical applications such as search-and-rescue, oil spill, and marine litter transport.

We thank the reviewer for recognising the relevance of our work and the very valuable detailed inputs. We have below carefully addressed all concerns raised below. Cursive text indicates direct quotes from the paper, while bold text represents new additions.

**General Comments:**

**Introduction (Section 1):**

The introduction emphasizes that the surface drifters used in this study represent buoyant objects better than other drifters. It sounds like a justification is not strictly necessary in such detail. (However, if comparative measurements with other drifter types exist, they should be mentioned here).

We agree with the reviewer that such level of detail in the comparison with other drifter types is not necessary. The intention was rather to convey that a key motivation for revisiting the topic of the relative importance of near-surface physical mechanisms in the transport of surface drifters is that these drifters have a novel design that distinguishes them from the rest. We have modified line 36-44 of the original manuscript to convey the topic more succinctly as follows:

*"Buoyant drifters can be used to monitor how these different physical mechanisms combine to drive the transport at the ocean near-surface (Lumpkin et al., 2017). Currently, the most commonly deployed drifters* **are transported with a current that is effectively integrated over part of the water column (over the vertical extent of the drifter) and do not capture directly the complex surface ocean dynamics**. *Alternatively, the surface drifters used in this study (see (MetOcean, 2020) ) have a thin disc shape* **with a height of 4.1 cm** *that enables them to follow the orbital velocities of the waves and drift with the uppermost* **centimeters of the ocean** *surface currents (Elipot et al., 2016), which are characterised by higher speeds. Yet, the relevance of the Stokes drift, wind drag, or surface currents on these specific drifters is currently unclear."*

**Specification of model data (Section 2.2):**

Building on Point 1: Since the drifter observations are later compared with predictions derived from model data using machine learning methods, it is more important to clarify the characteristics of the model data. From what depths do the model values originate? Are similar/identical depths being compared between the model and the drifter? Referring to "surface" is too general.

We thank the reviewer for this comment.

In line 112 we have added:
**"The model uses a hybrid z\*-σ terrain-following vertical coordinate system consisting of 51 levels with the thickness of the surface cell set to ≤1m (Tonani et al., 2019)**."

We have also clarified the impact of this finite upper layer on the definition of the Stokes drift on lines 124-125:
"As in (Bruciaferri et al., 2021), we calculate the Stokes drift at 0.5 m below the still–water level to align with the **depth mid-point** of the upper ocean model layer ..."

Further evaluation of the comparison of the surface velocity from this model and the drifter velocity has been included in the discussion section following the reviewer's suggestion. See our reply to the reviewer's comment on Discussion and Conclusion section.

**Variable naming (Section 2 and 3):**

It would help readability if variables were mentioned more in the text, but with a coherent/consistent approach (e.g. line 98 and lines 141/142).

We thank the reviewer for this suggestion, as we are committed to improve the readability of the paper. We have made the following changes to the method and results sections, where adding variable notation in the text would ease connecting the information to the figures:

Line 98: "(…) we estimate the net drifter **velocity** $v_d$ over the dominant tidal cycle T."

Line 140-141: "We define the peak spectral period vector $\vec{T_p} = \left(T_p^x, T_p^y\right) = \left(T_p \sin \theta^{bulk}, T_p \cos \theta^{bulk}\right)$ using bulk wave direction $\theta^{bulk}$ **and, in doing so, assign a scalar to the two directions in an ad hoc fashion. Similarly, we compute three different wave height vectors, each associated with a specific wave partition (wind sea, first swell, and second swell). These are expressed as** $\vec{H_s}^i = \left(H_s^{i,x}, H_s^{i,y}\right) = (H_s^i \sin \theta^i, H_s^i \cos \theta^i)$**, where i denotes the wave partition, and each vector is calculated using the corresponding wave direction for that partition."**

Lines 149-150: "(…) we model the zonal $(U_d)$ and meridional $(V_d)$ drifter velocity components separately in both the linear and machine learning models (…)"

Lines 324-327: (see changes in the answer to the next section)

Lines 330-331: "(…) we also observe small nonlinearities in the ALE plot of the meridional high-pass ocean currents $V_o^{HP}$"

Lines 333-334: "The ALE plots for the zonal wind velocity $U_{10}$ show that the effect on the zonal total drifter velocity $U_d$ becomes constant at extremes of the distribution, resembling a sigmoid function (…)"

Line 335: *"Meridional wind $V_{10}$ exhibits a similar (...)"*

Line 353: *"We find that meridional Stokes drift $V_s$ and meridional wind $V_{10}$ are highly correlated..."*

For lines 323-327, 350, and 358-361, see changes in the answers to the next sections.

**Quantitative description of results (Sections 4.1.1/4.1.2):**

The results section, particularly when describing Figures 2 and 4, would benefit from including more numerical values. For example, in Section 4.1.1, values could illustrate the reversed roles (line 325), and it could be shown that wind is the dominant factor for the meridional component but of comparable magnitude to zonal contributions.

The reviewer is right about this remark. Adding the specific values of the RMSE increase from each variable facilitates the comparison between them and improves readability.

We have made the following changes to lines 323-327:

*"(…) show there is a prominent signature of the tidal current in the zonal direction for the prediction of the drifter zonal velocity $U_d$, represented by a high RMSE increase of $U_o^{HP}$ (* **$0.37 \, m/s$ in random forest, $0.93 \, m/s$ in support vector regression),** *followed by a smaller contribution from the zonal wind $U_{10}$ ($0.14 \, m/s$, $0.23 \, m/s$). The roles of these two variables are reversed for the prediction of the meridional drifter velocity $V_d$, where models show a higher dependence on the meridional wind $V_{10}$ ($0.15 \, m/s$, $0.43 \, m/s$)* **compared to the contribution from the** *meridional high-pass ocean currents $V_o^{HP}$ ($0.05 \, m/s$, $0.13 \, m/s$)."*

We have also modified lines 350-351 as follows:
*"In the meridional direction, the wind $V_{10}$* **(RMSE increase of $0.10 \, m/s$ in random forest, $0.45 \, m/s$ in support vector regression)** *and the low-pass current $V_o^{LP}$ ($0.07 \, m/s$, $0.30 \, m/s$ respectively) are the most important features in both models.* **Yet, the** *meridional Stokes drift $V_s$ is only important in the random forest model* **compared the support vector regression model ($0.05 \, m/s$, $0.01 \, m/s$)."**

We refer the Reviewer #2 to the next comment on the Residual zonal velocity to see the rest of the changes.

**Residual zonal velocity (Section 4.1.2):**

Why is the residual zonal velocity only described in relation to the total velocity model? Although the effect is smaller, it should be reported for completeness.

The reviewer has a point here. Even though a general description is given in lines 348-349, we later refer specifically to the results of $\widetilde{V}_d$ but we do not further elaborate about $\widetilde{U}_d$.

We have modified lines 358-361 to clarify the results from the zonal residual model and the fact that the cross-terms are found for both models:

**"In the zonal direction, the wind $U_{10}$ (RMSE increase of $0.11 \, m/s$, $0.53 \, m/s$) and the Stokes drift $U_s$ ($0.06 \, m/s$, $0.18 \, m/s$) are the most important features in both models. The contribution of the low-pass ocean currents $U_o^{LP}$ is also comparable ($0.06 \, m/s$, $0.11 \, m/s$).**

**From both the residual zonal and meridional drifter velocity models,** *we also find high permutation feature importance of* **variables that are not aligned with the velocity component,**

*such as $V_{10}$ for $\widetilde{U}_d$ **(RMSE increase of** $0.02\ m/s,$ $0.09\ m/s$**)** and $H_s^{1st\ swell,x}$ for $\widetilde{V}_d$ **(** $0.01\ m/s,$* ***$0.09\ m/s$).***

**Discussion and conclusions (Sections 4 and 5):**

I support Reviewer #1's point that the discussion and conclusions are somewhat too short. Even the Random Forest model shows inaccuracies, which are small but should still be acknowledged and discussed. The model data used in Section 2.2 could also be critically assessed. Furthermore, the generalizability of the trained models to other regions should be addressed more explicitly. What aspects are transferable, and which are not? In general, the discussion would benefit from more detail as well as a stronger connection to the existing literature. As an addition to Reviewer #1: I also do not question the validity of the analysis method. However, if possible, the authors should consider including additional drifter data, either from their own campaigns or from other studies, for example Deyle et al. (2024) (if possible, with surface measurements from 0.5 m).

We appreciate the reviewer's comment highlighting how the discussion section could be strengthened.

1. Inaccuracies of machine learning models.
   The errors observed in the prediction of the machine learning models can arise from biases or noise in the training data, or from the model's inability to fully capture the underlying data structure (Molnar, 2022).

   We have addressed the first issue more explicitly by adding after line 387: ***"Deviations between the predicted and observed trajectories likely arise from biases in the training data caused by strong spatial correlations. As a result, (minor) velocity prediction errors can accumulate during integration, causing the simulated trajectories to diverge into regions not adequately represented in the training set."***

   We have also described in more detail the shortcomings of each machine learning model that might prevent them to capture the data structure:

   - In lines 190-192 regarding the random forest model, we have added:
     *"In this algorithm, each tree in the algorithm is fitted to a randomly drawn subset of the data and considers a random subset of features at each split. **Single decision trees struggle with linear relationships, which must be approximated by step functions, and are sensitive to small input changes, sometimes producing non-smooth predictions (Molnar, 2022). By averaging over many trees, random forests reduce these limitations,** exhibiting low sensitivity to hyperparameters such as the number of trees …."*

   - We have modified lines 202-205 regarding the support vector regression:
     *"This model applies a transformation using non-linear kernel functions to project the data into a higher-dimensional space where a linear hyperplane can better approximate the non-linear relationships within the data. **However, the support vector regression model is sensitive to the choice of kernel and hyperparameters, which can strongly affect model performance. To address this,** we use a radial basis function (RBF) and optimise the hyperparameters via grid search with cross-validation (see Appendix D1 for exact values)".*

2. Use of model data

As highlighted by Reviewer #2, using model-derived data to explain drifter motion introduces uncertainty, since these datasets do not perfectly represent real ocean conditions. This limitation may affect the accuracy of the training dataset and, consequently, the machine learning model's performance. Nevertheless, several factors ensure that the model fields employed here closely approximate real conditions in the open ocean.

- The use of hydrodynamic **reanalysis products,** which assimilate altimeter data, in situ temperature and salinity vertical profiles and satellite sea surface temperature provides a more accurate and realistic representation of the waves and currents.
- The use of **instantaneous current velocities** instead of hourly averages to match the real-time conditions. Using hourly averages would smooth out these fluctuations, effectively removing some of the variability that directly influences the drifter motion.
- The use of **linear interpolation** within Parcels to estimate at the exact positions and times of the drifters, which do not necessarily coincide with the discrete grid points or time steps of the model output.

Together, these factors make the model-based estimates a reliable approximation for interpreting the observed drifter dynamics at this spatiotemporal scale. A further comparison of the performance of this specific model for Lagrangian analysis can be found in Moerman et al. (2024).

However, we agree with Reviewer #2 that the uncertainties introduced by using model data should be acknowledged in the text and linked to existing literature, especially regarding the limited vertical resolution of the hydrodynamic models and its impact in the linear regression model. We have modified lines 175-178:

*"Nevertheless, despite the high interpretability of linear models, they may still oversimplify near-surface ocean dynamics by omitting non-linear behaviour in the parameterisation of drifter velocity components. For the case of highly correlated features, linear models also struggle to determine their contributions, leading to instability in coefficient estimation and reduced model reliability [3]. **From a physical perspective, another limitation is that the surface current velocities used here are depth averaged over the model's upper layer . This means that wind-driven vertical shear in the upper centimetres may be underestimated and part of the shear effect effectively absorbed by the windage coefficient, potentially influencing the interpretation of the modelled surface currents (Callies et al., 2019; Laxague et al., 2018).***

3. Transferability of results to other ocean regions

On the question of whether these results are transferable to other regions, we completely agree with the reviewer that this is an important consideration. We refer Reviewer #2 to our detailed response to Reviewer #1, who raised a similar concern.

4. Connection to background literature

We have ensured that the analysis of result transferability to other ocean regions includes a comparison with the predictive performance reported in other studies, providing context for the forecasting capabilities of state-of-the-art models.

We also agree that the discussion of the functional form of the wind contribution in the linear regression model would benefit from some more information about the background literature.

We have added after line 167 in the methodology:

**"Although this method finds an empirical parametrisation of the wind contribution, its functional form aligns with theoretically derived models of the drift of spherical buoyant objects at the ocean surface (Beron-Vera et al., 2019)"**

And after line 421 in the discussion:
**"*The linear parameterisation of the wind contribution to drifter velocities has been found to resemble the functional form derived by theoretical studies using the Maxey-Riley framework for buoyant spherical (Beron-Vera et al., 2019) and non-spherical (Wagner et al., 2022) particles. However, as noted by (Bos et al., in preparation), the particle Reynolds numbers for the surface drifters used in this study in the southern North Sea are above the Stokes drag regime. Therefore, it is important to consider that air and water have different viscosities when determining the functional form of the wind contribution to drifter velocity, which may no longer be linear*"**.

5. Short and too general conclusion
We understand and agree with the issue as raised by the reviewer. We have rewritten and extended the conclusion to connect back to the main goals of the study specified in lines 65-66. We have also added the new insights from the extra-analysis on the generalisability of the models to other regions:

*"This study analyses the effect of near-surface ocean currents, wave-induced motions, and wind drag on the trajectories of ultra-thin surface drifters to gain insight into the transport of buoyant objects at the ocean surface. We follow a data-driven approach and regress drifter velocity against instantaneous hydrodynamic and atmospheric conditions and compare the established linear leeway model with two fundamentally different machine learning **algorithms in terms of both inference of predominant forcing mechanisms and trajectory prediction performance.***

***At first order, machine learning models indicate that** wind and ocean currents are linearly related to drifter velocity **and are the most important features explaining the variability in velocity. These findings align with previous observational studies using leeway formulations (Breivik et al., 2011; Dominicis et al., 2016) and theoretical predictions from the Maxey-Riley framework (Beron-Vera et al., 2019). Stokes drift also contributes notably for low values of the residual (non-tidal) drifter velocity.***

*Non-linear behaviour emerges under strong wind, as revealed by the ALE plots of the drifter velocity (Fig. 3b**). Incorporating this insight, we improve trajectory predictions by using a quasi-linear model with a non-linear wind term. While linear approaches remain advantageous in operational oceanography due to their simplicity and computational efficiency, we propose a hybrid framework that utilises interpretable machine learning methods to reveal functional relationships between drifter velocity and environmental forcing. These insights can guide the formulation of more accurate linear parameterisations.***

***When evaluating trajectory predictions from integrated modelled** zonal and meridional velocities, the linear model performs reasonably well for predictions of less than 4 days but accumulates bias over longer periods. **In contrast,** machine learning models, especially the random forest, consistently outperform the linear baseline **(Fig. 8). This suggests that more complex models might be needed to extend the forecasting horizon.***

***Feature engineering analysis shows that** incorporating additional **physically relevant** information such as water depth or a parameterisation of wave-surfing effects, **further*

*improves random forest performance. Meanwhile, adding non-physical features, such as longitude and latitude, degrades predictions. Finally, we test the generalisability of the models to a region with markedly different ocean dynamics, the Tyrrhenian Sea. Performing 48h-reconstructions of surface drifter trajectories demonstrates a predictive skill comparable to other state-of-the-art studies, indicating low overfitting to the North Sea training data and reinforcing the physical conclusions of the near-surface ocean dynamics of this study."*

**Error estimation (Appendix A):**

I consider the error estimation to be problematic. With an interval of 2 hours, to my understanding, only 24 position values per drifter are available, which is insufficient for a robust error analysis. Are there any other studies available? This would strengthen this part.

We agree that only 24 data points per drifter might not be sufficient to analyse the accuracy of the GPS measurements. We have conducted an additional analysis placing 24 drifters of the same model used in the study in a static location on land to increase the number of datapoints to analyse. Two different measurement rounds were taken: a 24h cycle with a transmission frequency of 30 minutes, and a 3h measurement cycle with frequency of 5 minutes. In total, we analyse 84 position values for each drifter.

We have updated Figures A1.a) and A1.b) with the new data and have modified the text as follows:
*"We estimate the uncertainty in the spatial coordinates reported by the GPS system in the drifters from the errors in distribution of measurements during an experiment.* **We position Stokes drifters, identical to the ones used in this study, over a flat surface on land. A total of 84 coordinate data points were measured from each drifter, derived from two distinct measurement rounds. The first round comprised a 24-hour cycle with a 30-minute transmission frequency, and the second was a 3-hour cycle utilising a 5-minute frequency** *(Schneither & van Sebille, 2023). For each drifter, we compute the deviation of the longitude and latitude measurements with respect to their mean during the combined measuring periods and approximate their density distribution to continuous using the Kernel Density Estimation method. The resulting curves can be observed in Fig. A1, which highlights the mean standard deviation of the measurement distribution across drifters: 8.4 m in the latitudinal direction, and 6.5 m in the longitudinal direction."*

The caption from Figure A1. has also been updated accordingly:
*"Distribution of the deviation of (a) longitude and (b) latitude measurements with respect to their mean from* **24 colour-coded stationary surface drifters. Data were collected during a 24-hour cycle (30-minute frequency) and a 3-hour cycle (5-minute frequency).** *The distributions are estimated using Kernel Density Estimation (KDE). The mean standard error across all drifter distributions is included in the legend."*

**Consistency of text and figure (Appendix B):**

The description in the text does not fully match the figure. The M4 signal is visible in the FFT/PSD as well, not only in the Morlet Wavelet graph. indicates that intervals of 5 min, 30 min, and 3 h are considered, but the figure caption suggests just 3 h. It would also be helpful to indicate the 30 min boundary directly in the Morlet Wavelet graph to support the discussion in the text.

The reviewer is right about this inconsistency. The figure shows the average FFT and Morlet Wavelet analysis over all drifter time series after resampling to a 3hr interval.

For a sampling period of 3hr, the Nyquist period is 6hr, which closely matches the period of the M4 signal (6.2 hr). Hence, the period where the sampling frequency from the drifters is 3hr (day 26 onwards), the time resolution of the observations is barely enough to detect this signal.
However, resampling the data from 5 and 30 minutes to 3 hours manages to capture some of this signal that sits in the cutoff, but appears strongly damped.

To avoid this damping effect and effectively show the M4 signal in the data, we have computed the FFT and Morlet Wavelet analysis to two periods of time independently: from day 6-26 when the sampling period is 30 minutes, and from day 26 onwards when the sampling period is 3 hours. The analysis of the 5-minute period (day 1 to day 6) has not been included as the time interval between samples is more irregular (not exactly 300 s, with an average deviation of ±22 s). This irregular sampling complicates the alignment and averaging of results across drifters. The corresponding text in the manuscript has been updated.

Hence, Figure B1 has also been modified. The FFT graph shows 2 distinct lines representing the results from each of the sampling periods with their respective x-axis, and the Morlet Wavelet graph is a concatenation of two different PSD fields separated by a discontinued line. The colours of the tidal signals have also been modified to for accessibility for readers with colour vision deficiencies.

[Figure]

**Minor Comments:**
**Line 42**: Specify what is meant by "uppermost" already here (as done later in line 71).
Changed *"uppermost surface currents"* to *"uppermost **centimetres** of the ocean surface currents"*

Lines 84/85: Please clarify how time differences of 2.5 minutes can occur if the measurement interval is at least 5 minutes (line 81).
Lines 84-85 have been updated to:
*"To ensure the quality of the GNSS data, data points for which the time difference between measurements dt is less than 2.5min are eliminated. **This threshold removes highly measurements, which typically result from minor time synchronization errors between the drifter's internal time and the satellite time stamp or re-transmissions occurring within the intended sampling interval. Furthermore, the atmospheric and ocean data to be used for comparison (Sect. 2.2) is only available on a much coarser time resolution."***

**Lines 121/122**: The difference between the mean wave direction and the bulk wave direction should be explained more clearly.

We have changed lines 121-123: "*These parameters are derived from the wave spectrum model output, which partitions the spectral significant wave height ($H_s$) and the mean wave direction ($\theta$) into wind waves, and first and second swell components.* **The mean wave direction represents the energy-weighted average propagation direction within each spectral partition. In contrast, the bulk wave direction ($\theta_{bulk}$) is computed from the full, unpartitioned directional spectrum and represents the overall propagation direction of the total wave field.** *Additionally, we consider the period at the spectral peak ($T_p$) and the Stokes drift velocity field at the surface ($u_s$)."*

**Lines 325–327**: The comparison with the linear model is not sufficiently clear, please clarify.
We have changed the text accordingly to clarify the message. See changes in our answer to comments in section "Quantitative description of results" above.

**Line 397**: Please explain why including lat/lon as predictors decreased performance.

Latitude and longitude are just coordinates and they don't inherently carry physical meaning about the processes driving the target variable (i.e., drifter velocity). Hence, unless there exist specific stationary flow features in the underlying field driving the motion of the drifters, these features do not reflect any relationship between the physical variables and the target, and we do not expect them to improve the prediction.

We have further clarified this changing line 397: "**As expected**, *adding spatial coordinates such as latitude and longitude does not enhance model performance* **because the absolute location is not a relevant property, as it does not carry any physical meaning in a non-stationary flow**".

**Figure 9**: Caption contains an incorrect color reference for lat/lon, please correct.
Changed *"light blue"* to *"magenta"*

**REFERENCES**

Beron-Vera, F. J., Olascoaga, M. J., & Miron, P. (2019). Building a Maxey–Riley framework for surface ocean inertial particle dynamics. *Physics of Fluids*, *31*(9), 096602. https://doi.org/10.1063/1.5110731

Bos, M., Rypina, I. I., Pratt, L., & Van Sebille, E. (n.d.). *The Maxey-Riley-Gatignol equations for macroplastics in the North West European Shelf region*.

Breivik, Ø., Allen, A. A., Maisondieu, C., & Roth, J. C. (2011). Wind-induced drift of objects at sea: The leeway field method. *Applied Ocean Research*, *33*(2), 100–109. https://doi.org/10.1016/j.apor.2011.01.005

Bruciaferri, D., Tonani, M., Lewis, H., Siddorn, J., Saulter, A., Castillo, J. M., Garcia Valiente, N., Conley, D., Sykes, P., Ascione, I., & McConnell, N. (2021). The Impact of Ocean-Wave Coupling on the Upper Ocean Circulation During Storm Events. *Journal of Geophysical Research, Oceans*, *126*(6). https://doi.org/10.1029/2021JC017343

Callies, U., Carrasco, R., Floeter, J., Horstmann, J., & Quante, M. (2019). Submesoscale dispersion of surface drifters in a coastal sea near offshore wind farms. *Ocean Science*, *15*(4), 865–889. https://doi.org/10.5194/os-15-865-2019

Calvert, R., McAllister, M. L., Whittaker, C., Raby, A., Borthwick, A. G. L., & Van Den Bremer, T. S. (2021). A mechanism for the increased wave-induced drift of floating marine litter. *Journal of Fluid Mechanics*, *915*, A73. https://doi.org/10.1017/jfm.2021.72

Dominicis, M. D., Bruciaferri, D., Gerin, R., Pinardi, N., Poulain, P. M., Garreau, P., Zodiatis, G., Perivoliotis, L., Fazioli, L., Sorgente, R., & Manganiello, C. (2016). A multi-model assessment of the impact of currents, waves and wind in modelling surface drifters and oil spill. *Deep Sea Research Part II: Topical Studies in Oceanography*, *133*, 21–38. https://doi.org/10.1016/j.dsr2.2016.04.002

Laxague, N. J. M., Özgökmen, T. M., Haus, B. K., Novelli, G., Shcherbina, A., Sutherland, P., Guigand, C. M., Lund, B., Mehta, S., Alday, M., & Molemaker, J. (2018). Relative dispersion at hte surface of the Gulf of Mexico. *Geophysical Research Letters*, *45*(1), 245–249. https://doi.org/10.1002/2017GL075891

Lumpkin, R., Özgökmen, T., & Centurioni, L. (2017). Advances in the Application of Surface Drifters. *Annual Review of Marine Science*, *9*(Volume 9, 2017), 59–81. https://doi.org/10.1146/annurev-marine-010816-060641

MetOcean. (2020). *Stokes Drifter*. MetOcean.

Moerman, B., Breivik, Ø., Hole, L. R., Hope, G., Johannessen, J. A., & Rabault, J. (2024). *An analysis on OpenMetBuoy-v2021 drifter in-situ data and Lagrangian trajectory simulations in the Agulhas Current System* (Version 1). arXiv. https://doi.org/10.48550/ARXIV.2409.20096

Molnar, C. (2022). *Interpretable Machine Learning: A Guide for Making Black Box Models Explainable* (2nd ed.). https://christophm.github.io/interpretable-ml-book/. https://christophm.github.io/interpretable-ml-book

Morey, S. L., Wienders, N., Dukhovskoy, D. S., & Bourassa, M. A. (2018). Measurement Characteristics of Near-Surface Currents from Ultra-Thin Drifters, Drogued Drifters, and HF Radar. *Remote Sensing*, *10*(10), 1633. https://doi.org/10.3390/rs10101633

Pawlowicz, R., Chavanne, C., & Dumont, D. (2024). *The Water-Following Performance of Various Lagrangian Surface Drifters Measured in a Dye Release Experiment*. https://doi.org/10.1175/JTECH-D-23-0073.1

Schneither, M., & van Sebille, E. (2023). *Stokesdrifters_Wadden*. https://github.com/Parcels-code/Stokesdrifters_Wadden

Tonani, M., Sykes, P., King, R. R., McConnell, N., Péquignet, A.-C., O'Dea, E., Graham, J. A., Polton, J., & Siddorn, J. (2019). The impact of a new high-resolution ocean model on the Met Office North-West European Shelf forecasting system. *Ocean Science*, *15*, 1133–1158. https://doi.org/10.5194/os-15-1133-2019

Wagner, T. J. W., Eisenman, I., Ceroli, A. M., & Constantinou, N. C. (2022). *How Winds and Ocean Currents Influence the Drift of Floating Objects*. https://doi.org/10.1175/JPO-D-20-0275.1